# Characterizing the interactions between influenza and respiratory syncytial viruses and their implications for epidemic control

Sarah C. Kramer [1] ✉, Sarah Pirikahu[1], Jean-Sébastien Casalegno[2,3,4] & Matthieu Domenech de Cellès [1]

Pathogen-pathogen interactions represent a critical but little-understood feature of infectious disease dynamics. In particular, experimental evidence suggests that influenza virus and respiratory syncytial virus (RSV) compete with each other, such that infection with one confers temporary protection against the other. However, such interactions are challenging to study using common epidemiologic methods. Here, we use a mathematical modeling approach, in conjunction with detailed surveillance data from Hong Kong and Canada, to infer the strength and duration of the interaction between influenza and RSV. Based on our estimates, we further utilize our model to evaluate the potential conflicting effects of live attenuated influenza vaccines (LAIV) on RSV burden. We find evidence of a moderate to strong, negative, bidirectional interaction, such that infection with either virus yields 40-100% protection against infection with the other for one to five months. Assuming that LAIV reduces RSV susceptibility in a similar manner, we predict that the impact of such a vaccine at the population level would likely depend greatly on underlying viral circulation patterns. More broadly, we highlight the utility of mathematical models as a tool to characterize pathogen-pathogen interactions.

Although pathogens are often studied in isolation, mounting evidence suggests that interactions, in which one pathogen impacts the risk of infection or disease due to another pathogen, are a common and important feature governing their epidemiological dynamics[1–3]. These interactions can be characterized according to their major components: they may be positive (i.e., infection with one pathogen facilitates or exacerbates infection with another) or negative (i.e., infection with one pathogen inhibits or mitigates infection with another); may have a strong or a weak effect; may be short-lived or long-lasting; may be symmetric (i.e., both pathogens impact each other in the same way) or asymmetric; and may operate according to a range of biological mechanisms, including changes to the immune response or to host-cell gene expression[4].

The mechanisms underlying these interactions often occur at the individual level, but the existence of an interaction can lead to significant, and sometimes unexpected, consequences at the population level[5,6]. For example, if infection with one pathogen increases susceptibility to another, as demonstrated for influenza viruses and the *Streptococcus pneumoniae* bacterium[7,8], public health measures designed to reduce infection with the first pathogen may also succeed in combating the cocirculating pathogen. Conversely, if infection with one pathogen reduces susceptibility to

[1]Max Planck Institute for Infection Biology, Infectious Disease Epidemiology group, Charitéplatz 1, Campus Charité Mitte, 10117 Berlin, Germany. [2]Hospices Civils de Lyon, Hôpital de la Croix-Rousse, Centre de Biologie Nord, Institut des Agents Infectieux, Laboratoire de Virologie, Lyon, France. [3]Centre national de référence des virus des infections respiratoires (dont la grippe), Hôpital de la Croix-Rousse, Lyon, France. [4]Centre International de Recherche en Infectiologie (CIRI), Laboratoire de Virologie et Pathologie Humaine - VirPath Team, INSERM U1111, CNRS UMR5308, École Normale Supérieure de Lyon, Lyon, France. ✉e-mail: kramer@mpiib-berlin.mpg.de

another, then measures to control this pathogen may lead to unintentional increases in the burden of the other. For example, seasonal influenza vaccination has been associated with an increased risk of both pandemic influenza H1N1 2009 (H1N1pdm09)[9] and non-influenza respiratory virus infection[10], a finding that could be due to the existence of negative interactions. For this reason, it is critical that interactions between pathogens be understood and taken into account when developing public health policy.

One such interaction of public health relevance may exist between influenza and respiratory syncytial viruses (RSV), both of which represent significant public health threats, particularly in young children and the elderly[11]. Experimental studies have found that ferrets and mice infected with influenza were less likely to be infected with RSV upon challenge[12,13], and that infection with either of the two viruses can reduce morbidity upon subsequent infection with the other[12,14,15]. In human populations, this interaction is tentatively supported by epidemiologic observations revealing delays in RSV epidemics occurring during and immediately after the 2009 influenza pandemic[16,17]. Evidence to date is far from conclusive; studies on the severity of influenza-RSV coinfections, in particular, often come to conflicting results, with some studies suggesting that the interaction instead leads to increased disease severity[18,19]. A better understanding of this interaction may play a key role in informing public health practice. In the presence of a negative interaction, preventing influenza cases through widespread vaccination may lead to an unintentional increase in RSV burden. However, if vaccination with a live attenuated influenza vaccine (LAIV) induces an immune response similar to natural infection, widespread use of this vaccine, in particular, could help prevent RSV transmission. Understanding which of these conflicting effects is likely to dominate will help medical and public health practitioners alike better prepare for the RSV season.

Epidemiologic evidence alone is insufficient to confirm the existence of this interaction. Observable metrics such as phase differences (i.e., the difference in timing between two outbreaks) and the prevalence of coinfections are prone to mischaracterizing both the strength and direction of interactions[6,20]. The applicability of studies in animal models to human populations is also unclear. Mathematical modeling approaches offer a promising alternative to standard epidemiologic study designs because they are able to capture the complex and nonlinear dynamics inherent to infectious disease transmission[21], and to explicitly account for the mechanisms underlying an interaction[5,20,22]. By confronting these models with data, it is possible to understand the role these properties may play in generating the outbreak patterns we observe in reality. Mathematical models are also well-suited to exploring the impacts of prospective public health control measures, as they can be used to generate counterfactual scenarios[5]. Despite this potential, to our knowledge, only one modeling study has attempted to infer the characteristics of the interaction between influenza and RSV, and found that data were equally consistent with either a moderate negative interaction or no interaction[23].

Here, we fit a mathematical model of influenza and RSV cocirculation to multiple seasons of data from two locations, Hong Kong and Canada, in order to infer the strength and duration of the interaction between RSV and different (sub)types of influenza at the individual level. Additionally, we use our model to assess the potential impact of widespread LAIV use on the burden of RSV. We find evidence that infection with influenza or RSV moderately or strongly reduces susceptibility to the other virus, and that this reduction may persist for up to several months. Furthermore, we demonstrate that this interaction may, in some cases, play a substantial role at the population level. Finally, we show that the impact of LAIV on RSV burden is likely to be highly dependent on the circulation patterns of both viruses.

## Results

### Influenza and RSV epidemic patterns are consistent from year to year

We analyzed weekly data on (1) the number and rate of samples testing positive for influenza and RSV and (2) the proportion of influenza-like illness (ILI) per all-cause consultation in Hong Kong and Canada over several years. Because testing was primarily performed on samples taken from hospitalized and emergency department patients, these data are likely biased toward the most severe cases. To achieve a more representative picture of influenza and RSV circulation among the population as a whole, we incorporated ILI data as well. Despite Hong Kong's subtropical climate, epidemics of influenza A(H1N1) and B typically followed a clear seasonal pattern, with a single peak occurring during the winter or spring of each year (Fig. 1a and Supplementary Fig. 2). Thus, we summed cases of influenza A(H1N1) and B for our main analyses in Hong Kong. Due to the presence of multi-peak epidemics, which are challenging to describe using simple models, we did not consider interactions with influenza A(H3N2) (Supplementary Fig. 2), although supplementary analyses attempt to account for its circulation (see Supplementary Text). In Canada, where outbreaks of all influenza subtypes display a clear seasonal pattern, we summed all influenza cases for our analyses (Fig. 1b).

In Hong Kong, epidemics of RSV displayed weaker seasonality than those of influenza, with less definite peaks and more persistent circulation throughout the year. This was not the case in Canada, where outbreaks of both viruses exhibited similar durations and noisiness. There was considerable overlap of influenza and RSV activity during most outbreaks across all locations, although peak activity for influenza almost always preceded peak activity for RSV (median difference = 20 weeks in Hong Kong, 8.5 weeks in Canada). We observed no clear correlation between time series of influenza and RSV positivity rates ($p = 0.08$, Kendall's $\tau = -0.067$, 95% confidence interval $-0.14$–$0.008$) in Hong Kong, and a moderate positive correlation in Canada ($p < 1e-10$, Kendall's $\tau = 0.62$, 95% confidence interval = $0.57$–$0.66$). However, as discussed in the Introduction, such correlations alone are uninformative about the underlying interaction between the two viruses. The total percentage of samples testing positive for each virus was higher in Canada than in Hong Kong. Across both locations, season-to-season attack rates were more variable for influenza than for RSV (Supplementary Table 1).

Peak ILI activity often coincided with influenza and RSV activity, although the ILI data were typically noisier (Fig. 1c). This is likely due to ILI being a nonspecific, symptomatic measure that can be caused by a wide array of pathogens.

### There exists a negative, bidirectional interaction between influenza and RSV, with a duration of up to several months

In order to characterize the strength and duration of the interaction between influenza and RSV, we fit a deterministic, compartmental model of influenza and RSV cocirculation (Fig. 2) to the data described above. We modeled the interaction effect as a decrease in the susceptibility to infection with one virus that occurs during infection with the other virus and that persists for some time after recovery. We fit the model to all available seasons for a given location simultaneously, such that parameters describing the interaction effect were constrained to take the same value in all seasons, whereas several season-specific parameters were allowed to vary. Confidence intervals were obtained using a parametric bootstrap[24]. A description of all model parameters can be found in Table 1, and full model equations can be found in Supplementary Equation (1). All model fitting was conducted using a maximum likelihood approach. More details on how models were fit can be found in the Methods and in the Supplementary Text.

Among those currently or recently infected with influenza, we found evidence of a reduction in susceptibility to RSV by almost

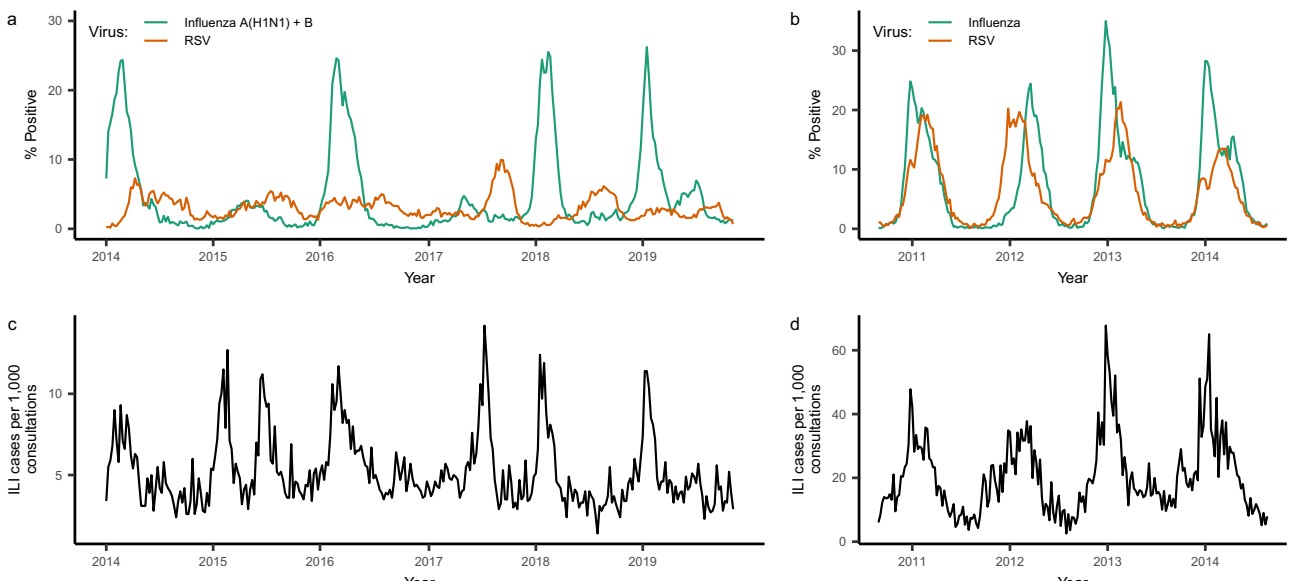

**Fig. 1 | Respiratory virus positivity and ILI cases over the study period in Hong Kong (a, c) and Canada (b, d). a**, **b** Weekly percentage of tests positive for influenza (A(H1N1) and B in Hong Kong, all subtypes in Canada) and RSV; **c**, **d** weekly number of cases of influenza-like illness reported by public out-patient clinics (Hong Kong) or primary care providers (Canada), per 1000 consultations. Detailed information on where to obtain these data can be found in the data and code availability statements.

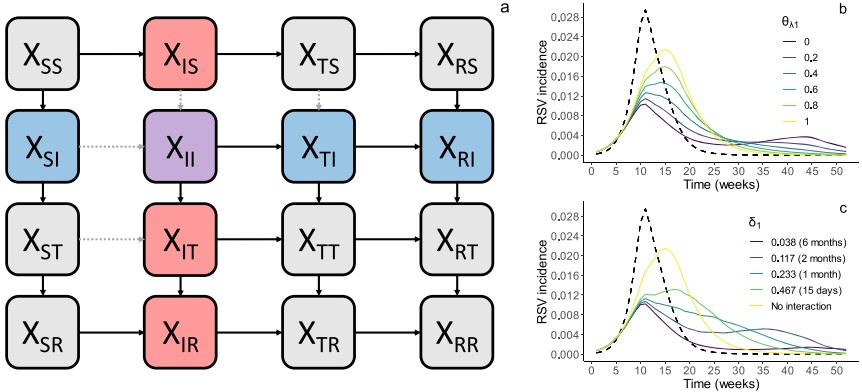

**Fig. 2 | Model of influenza and RSV cocirculation. a** Model schematic. Boxes represent the model states, where the first subscript indicates infection status with regard to influenza and the second subscript indicates infection status with regard to RSV (i.e., $X_{IR}$ refers to those infected with influenza and recovered from RSV). Compartments containing individuals infected with influenza are shown in red, compartments with individuals infected with RSV are shown in blue, and the compartment containing coinfected individuals is shown in purple. Arrows represent possible transitions between compartments; the gray dotted arrows, in particular, indicate transitions that are influenced by the interaction effect. **b**, **c** Impact of interaction parameters on outbreak trajectories. Plots show simulated RSV outbreaks from the model in (**a**) when varying values for the strength ($\theta_{\lambda I}$, **b**) and duration ($\delta_I$, **c**) of the interaction effect of influenza on RSV are modeled. The strongest (**b**) and longest-lasting (**c**) interaction effects are plotted in purple, while yellow lines show RSV outbreaks in the absence of an interaction. The corresponding influenza outbreak is plotted as a dotted line.

100% in Hong Kong ($\theta_{\lambda I} = 0$, 95% confidence interval = 0–0.0003), and by 39% in Canada ($\theta_{\lambda I} = 0.61$, 95% confidence interval = 0.51–0.62). The inferred duration of this reduction was roughly 100 days (95% confidence interval = 96–112 days) in Hong Kong and roughly 75 days (95% confidence interval = 32–144 days) in Canada (Table 1). We find comparable estimates for the strength of the effect of RSV on susceptibility to influenza ($\theta_{\lambda 2} = 0$, 95% confidence interval = 0–0.13 in Hong Kong; $\theta_{\lambda 2} = 0$, 95% confidence interval = 0–0.013 in Canada). However, the duration of this effect was considerably shorter than that of the effect of influenza on RSV (36 days, 95% confidence interval = 36–44 days in Hong Kong; 20 days, 95% confidence interval = 19–22 days in Canada).

To assess the role that these interactions play at the population level, we simulated epidemics at the maximum likelihood estimate (MLE) in the absence of an interaction effect. We predicted that RSV epidemics would be, on average, 27% larger (range across seasons: 7 to 40%) in Hong Kong and 5% larger (range: 0 to 11%) in Canada if influenza had no effect on RSV susceptibility. Meanwhile, attack rates of influenza would be, on average, 26% larger (range: 9 to 38%) in Hong Kong and 31% larger (range: 21 to 45%) in Canada if RSV had no impact on influenza susceptibility (Supplementary Fig. 12). Overall, our results suggest that the negative interaction between influenza and RSV can act to noticeably limit the yearly attack rates of both viruses, although this is dependent on the circulation patterns of both viruses.

**Table 1 | Descriptions and maximum likelihood estimates of all model parameters. Results are provided as a Source Data file**

| Parameter | Description | Fit Value (95% CI) | | Season-specific? |
|---|---|---|---|---|
| | | Hong Kong | Canada | |
| $\theta_{\lambda 1}$ | Strength of the interaction effect of influenza on RSV | $1.2 \times 10^{-9}$ (0, $3.2 \times 10^{-4}$) | 0.61 (0.51, 0.62) | No |
| $\theta_{\lambda 2}$ | Strength of the interaction effect of RSV on influenza | $6.2 \times 10^{-6}$ (0, 0.13) | 0 (0, 0.013) | No |
| $\delta_1$ | Rate of loss of cross-protection against RSV after influenza infection | 0.065 (0.063, 0.073) | 0.092 (0.049, 0.22) | No |
| $d_2$ | Rate of loss of cross-protection against influenza after RSV infection, relative to $\delta_1$ | 3.0 (2.3, 3.0) | 3.9 (1.3, 5.5) | No |
| $\rho_1$ | Composite reporting rate/scaling parameter for influenza | 0.20 (0.18, 0.20) | 2.7 (2.7, 3.2) | No |
| $\rho_2$ | Composite reporting rate/scaling parameter for RSV | 0.034 (0.032, 0.035) | 0.84 (0.79, 0.92) | No |
| $\eta_{temp1}$ | Impact of temperature on influenza | −0.15 (−0.15, −0.13) | NA | No |
| $\eta_{AH1}$ | Impact of absolute humidity on influenza | 0.23 (0.21, 0.23) | NA | No |
| $\eta_{temp2}$ | Impact of temperature on RSV | −0.089 (−0.12, −0.083) | NA | No |
| $\eta_{AH2}$ | Impact of absolute humidity on RSV | 0.19 (0.18, 0.22) | NA | No |
| $b_1$ | Amplitude of seasonal forcing for influenza | NA | 0.19 (0.18, 0.19) | No |
| $b_2$ | Amplitude of seasonal forcing for RSV | NA | 0.15 (0.14, 0.16) | No |
| $\varphi_1$ | Week of maximum seasonal forcing for influenza | NA | 26.2 (25.7, 26.5) | No |
| $\varphi_2$ | Week of maximum seasonal forcing for RSV | NA | 30.2 (29.0, 30.5) | No |
| $\alpha$ | Amplitude of seasonality in reporting and background consultation rates | 0.61 (0.57, 0.64) | 0.90 (0.90, 0.91) | No |
| $\varphi$ | Week of maximum $\rho_1/\rho_2$ | 43.8 (43.3, 44.3) | 48.0 (47.9, 48.2) | No |
| $Ri_1$ | Initial effective reproductive number for influenza | See Supplementary Fig. 4 | | Yes |
| $Ri_2$ | Initial effective reproductive number for RSV | See Supplementary Fig. 4 | | Yes |
| $I_{10}$ | Initial proportion of the population infected with influenza | See Supplementary Fig. 4 | | Yes |
| $I_{20}$ | Initial proportion of the population infected with RSV | See Supplementary Fig. 4 | | Yes |
| $R_{10}$ | Initial proportion of the population immune to influenza only | See Supplementary Fig. 4 | | Yes |
| $R_{20}$ | Initial proportion of the population immune to RSV only | See Supplementary Fig. 4 | | Yes |
| $R_{120}$ | Initial proportion of the population immune to both influenza and RSV | See Supplementary Fig. 4 | | Yes |

## Both temperature and absolute humidity modulate the dynamics of influenza and RSV

Optimal model fit data from Hong Kong was achieved when both temperature and absolute humidity were included in the model (Supplementary Table 5, tested for significance using a likelihood-ratio test), suggesting that temperature and absolute humidity both modulate the transmission of influenza and RSV. Specifically, our results suggest that transmission rates of both influenza and RSV are negatively influenced by temperature when controlling for absolute humidity, and positively influenced by absolute humidity when controlling for temperature (Table 1). The impact of temperature appears to be stronger for influenza than for RSV, while the impact of absolute humidity is similar for both viruses. The overall extent of seasonal forcing over time for both viruses is shown in Supplementary Fig. 8. Because of the large geographic range over which data from Canada were collected, we did not explicitly incorporate climate forcing in the model when fitting to data from Canada.

## Inferred reproductive numbers and immune fractions are similar across locations but highly variable between seasons

In addition to the parameters discussed above, we also fit several season-specific parameters, including the effective reproductive numbers ($R_{i1}$ for influenza and $R_{i2}$ for RSV, where $i$ indicates the season) and the proportion of the population immune to influenza ($R_{10} + R_{120}$) and RSV ($R_{20} + R_{120}$) at the beginning of each season (where $R_{10}$ indicates the proportion of the population immune to influenza alone, $R_{20}$ to RSV alone, and $R_{120}$ to both viruses) (Table 1 and Supplementary Fig. 3). The season-specific MLEs of the effective reproductive numbers for influenza ranged from 1.3 to 1.7 in Hong Kong, and from 1.3 to 1.6 in Canada. Estimates for RSV ranged from 1.2 to 2.0 in Hong Kong, and from 1.8 to 1.9 in Canada.

We find that immunity to influenza and RSV is highly dependent on the season. Specifically, the season-specific percentage of the

population immune to influenza at the beginning of a season ranged from 43 to 91% in Hong Kong, and from 39 to 83% in Canada. Values inferred for RSV were similarly variable but tended to be lower in magnitude, ranging from 15 to 56% of the model population in Hong Kong, and from 21 to 44% in Canada.

## The best-fitting models achieve high-quality fit to data on interacting pathogens

The predicted case counts simulated from the fitted model at the MLE are highly correlated with the observed data (Fig. 3 and Supplementary Fig. 9), indicating that the fit model is capable of accurately reproducing observed epidemic patterns. The quality of the model fit is comparable across viruses and locations, with the exception of RSV in Hong Kong, where model fit quality is substantially lower (Fig. 3b). This could be due to the weaker seasonality of these data (Fig. 1a and Supplementary Table 1), which can be more difficult for simple models like the one used here to capture. When comparing model fit to that of a sine wave fit to all data from a given location, our transmission model still fits substantially better except in the case of RSV in Canada, where our model fits similarly to the sine wave (Supplementary Table 3). Furthermore, we find that simulations at the MLE are generally able to accurately reproduce the peak timing, peak case counts, and overall attack rates observed each season (Supplementary Table 4). Collectively, these results support the utility of mathematical models as tools for describing the characteristics of pathogen–pathogen interactions.

## Sensitivity analyses demonstrate the robustness of the results

As described in the Methods below, we conducted multiple rounds of model fitting in order to improve our chances of achieving convergence to the MLE. To further ensure convergence, we calculated profile likelihoods[25] for $\theta_{\lambda 1}$ for both locations. The profile likelihoods peaked at or near the MLEs (Supplementary Fig. 10), demonstrating that our model-fitting approach was indeed able to reach the true MLE.

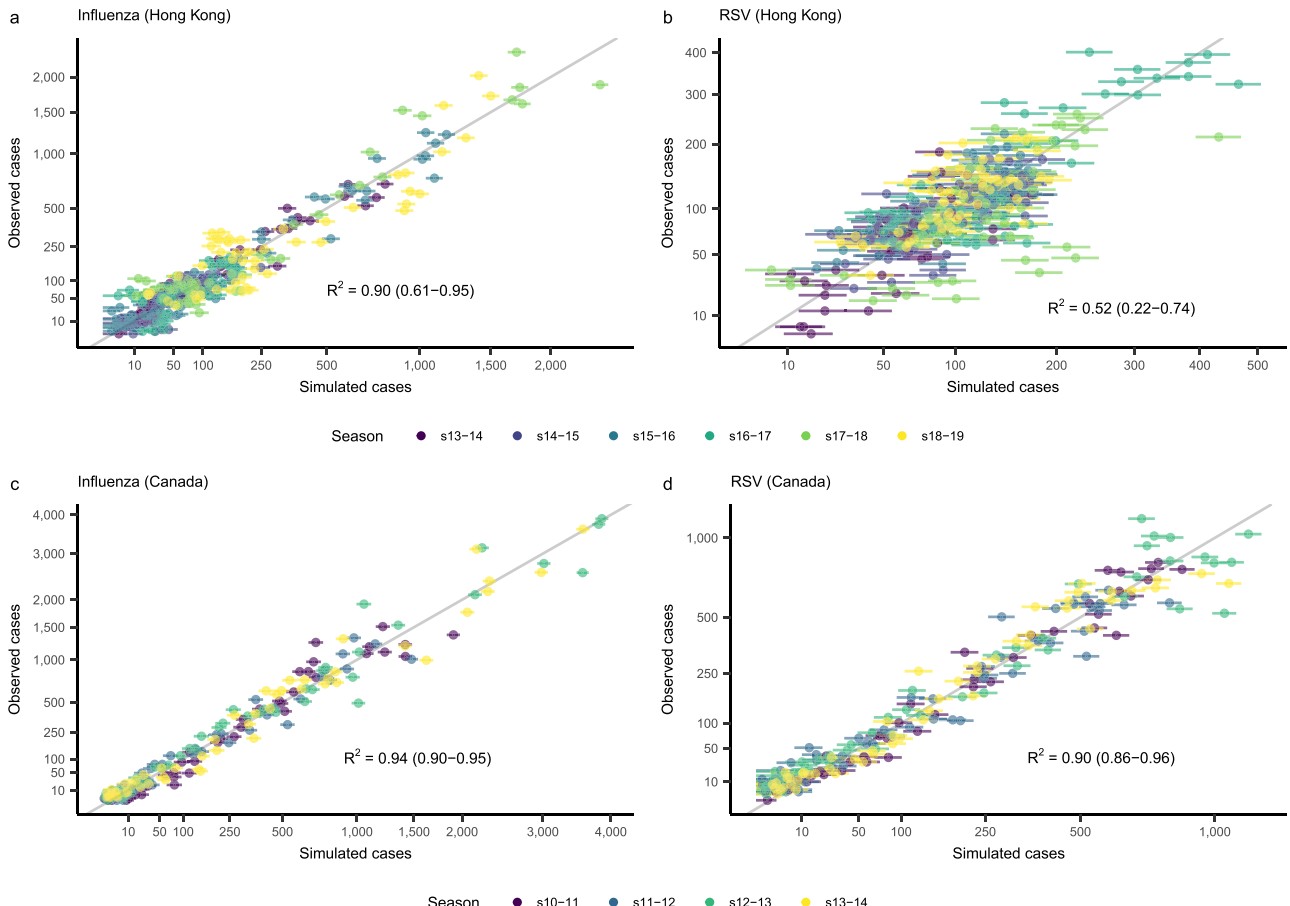

**Fig. 3 | Quality of model fit assessed at the maximum likelihood estimates of all parameters.** Cases simulated from the deterministic model at the MLE vs. observed case counts in Hong Kong (**a**, **b**) and Canada (**c**, **d**). More specifically, simulated case counts were taken to be the mean value of the observation model at each time point. Error bars represent the 95% prediction intervals, obtained for each point from a binomial distribution with a number of trials equal to the observed number of tests performed for influenza or RSV at that time point, and the probability of success equal to the model-defined probability of a positive test (Eq. 3). Colors indicate the season from which each point was taken. The gray line is the identity line, and represents perfect agreement between simulations and observations. Results are shown separately for influenza (**a**, **c**) and RSV (**b**, **d**). The Nash-Sutcliffe coefficient of efficiency ($R^2$)[95] values over all seasons combined, as well as the range of values by season, are shown. Note that both axes use a square root transformation. Detailed information on where to obtain the data shown here an be found in the Data and Code Availability Statements.

Because transmission patterns of both influenza and RSV are dependent on age[26–28], it is possible that an age-structured model would improve the estimation of interaction characteristics. We therefore conducted a sensitivity analysis in order to evaluate whether our model, which assumes homogeneous mixing, was capable of accurately inferring interaction parameters from data derived from an age-structured population. Briefly, we generated synthetic data for all seasons at the MLE for Hong Kong using an age-structured model, and fit our homogeneous mixing model to these data (see Supplementary Text). Fitted values of all shared parameters agreed closely with the values used to generate the data, suggesting that our model is capable of achieving the goals outlined in this work.

**Influenza vaccination may increase the burden of RSV, although this effect is highly dependent on underlying patterns of influenza and RSV circulation**
There is a wealth of evidence suggesting that some vaccines, especially live vaccines, can lead to temporary, nonspecific protection against non-target pathogens, likely through upregulation of the innate immune system[29,30]. Although the potential for LAIV to offer such cross-protection has not been widely assessed, one study demonstrated that LAIV was able to reduce RSV replication in mice[31], and several studies have shown upregulation of the innate immune system after vaccination

with LAIV[30]. If LAIV can indeed offer protection against RSV, we might expect widespread use of LAIV to reduce the population-level burden of RSV. On the other hand, given the negative interaction between influenza and RSV, a reduced burden of influenza due to widespread LAIV use could lead to inadvertent increases in the burden of RSV.

In order to assess which of these competing effects is expected to dominate during realistic epidemics, we simulated the effect of LAIV administration, assuming imperfect or "leaky" immunity[32], at several time points and coverage levels on RSV attack rates. Model equations including LAIV vaccination can be found in Supplementary Equation (2), and the model schematic in Supplementary Fig. 13. We modeled two scenarios: a "subtropical" scenario, where parameter values were set to their MLEs as estimated from the Hong Kong data, and a "temperate" scenario, where parameter values were set to their MLEs as estimated from the Canada data. The strength and duration of the interaction effect were set to the values obtained from Hong Kong for both scenarios. This was done to remove these parameters as potential confounders of the relationship between the underlying circulation patterns of both viruses and the impact of the vaccine on RSV burden, although we also conducted a sensitivity analysis using the interaction parameters obtained from Canada. We ran our analysis for two vaccines: one with a strong effect on RSV susceptibility (as found in Hong Kong), and one with a more moderate effect on RSV susceptibility (as found in Canada).

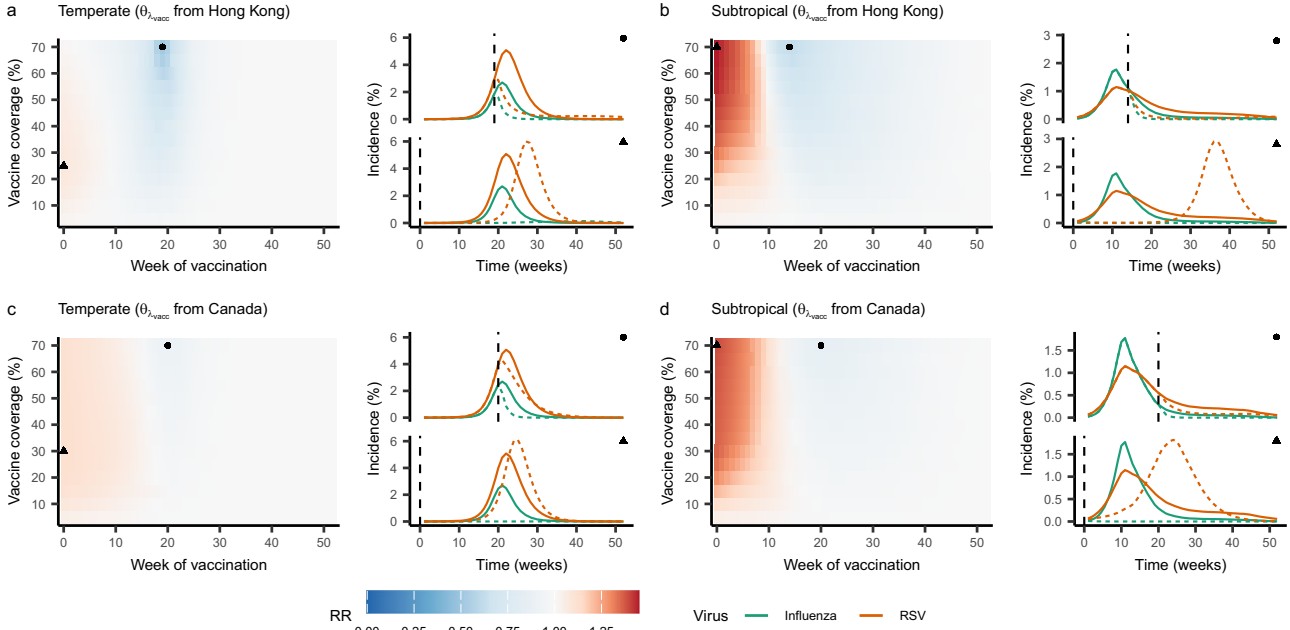

**Fig. 4 | Impact of vaccination with LAIV on RSV attack rates.** Heatmaps show the rate ratio of RSV attack rates in the presence of vaccination relative to the attack rates in the absence of vaccination (Eq. (5)) at a range of values for vaccine timing (x-axes) and coverage levels (y-axes). Values below 1.0 (indicating a decrease in attack rates) are shown in blue, and values above 1.0 (indicating an increase in attack rates) are shown in red. Lines indicate the simulated incidence of influenza (green) and RSV (orange) over the season in both the absence (solid lines) and presence (dotted lines) of vaccination. Points are used to indicate the timing and coverage level of vaccination for each simulation; specifically, circles indicate the combination of vaccine timing and efficacy that yielded the greatest reduction in RSV attack rates, while triangles indicate the combination yielding the greatest increase. A maximum vaccine coverage of 70% is shown, as coverage levels above this are unlikely to be realistic. Results are shown for both temperate (**a, c**) and subtropical (**b, d**) scenarios. Results assuming a strong impact of vaccination on RSV susceptibility, as found in Hong Kong, are shown in (**a, b**); results assuming a moderate impact of vaccination on RSV susceptibility, as found in Canada, are shown in (**c, d**). For each scenario, a representative season was selected (2010–11 for the temperate scenario and 2018–19 for the subtropical scenario); results for all other seasons can be found in Supplementary Fig. 14. Source data are provided as a Source Data file.

In both the temperate and subtropical scenarios, we found that early vaccination with LAIV typically led to increases in RSV attack rates, although this effect was, on average, weaker in the temperate scenario (Fig. 4 and Supplementary Fig. 14). The increase in RSV burden was most pronounced at vaccine coverage levels of 25-30% in the temperate scenario (Fig. 4a, c) and at coverage levels of 70% and above in the subtropical scenario (Fig. 4b, d). Reductions in RSV attack rates were only consistently observed if LAIV was delayed for several weeks, and if coverage levels were relatively high. In the case of a vaccine with a more moderate impact on RSV susceptibility (Fig. 4c, d), increases in RSV burden were typically larger and reductions smaller, although this pattern did not always hold in the subtropical scenario. Similar results were observed when the interaction parameters were instead set to those fit in Canada (Supplementary Fig. 15), and results were broadly consistent across a range of sensitivity analyses in which the strength and duration of the vaccine effect were varied (Supplementary Fig. 16). However, we note that the impact of LAIV on RSV burden was highly dependent on the timing and intensity of the underlying influenza and RSV outbreaks. In particular, when RSV outbreaks peaked before influenza outbreaks, and when RSV outbreaks were large relative to influenza outbreaks, LAIV more consistently led to decreases in RSV burden (Supplementary Fig. 14).

## Discussion

In this work, we develop a general mathematical model of the cocirculation of two viruses, and fit it to observed virologic and syndromic data to characterize the interaction between influenza viruses and RSV. We find evidence that infection with influenza provides moderate to strong protection against infection with RSV for up to 1–5 months postrecovery, and that RSV may have a strong but shorter-lived effect on susceptibility to influenza. Furthermore, we find that these individual-level effects can have substantial implications for population-level epidemic dynamics and for control by vaccination. Altogether, we provide one of the first estimates of the substantial individual- and population-level effects of the interaction between influenza and RSV, and highlight the utility of a mathematical modeling approach in characterizing virus-virus interactions.

Our estimates of the strength and direction of the impact of influenza infection on susceptibility to RSV are consistent with past experimental work[12,13]. Furthermore, our simulations in the absence of an interaction effect confirm that this interaction can have a substantial impact on RSV circulation at the population level, as possibly observed during the 2009 influenza pandemic[16,17]. Our estimate of the duration of this effect, however, was longer than expected. An experimental study in ferrets demonstrated that susceptibility to RSV returned to baseline levels soon after the clearance of influenza[12]. Similarly, a past experiment in mice suggested that the upregulation of proteins involved in the interferon response, which is typically short-lived, played an important role in driving the interaction effect[13]. In contrast, we found evidence of an interaction effect that persisted for several months post-infection. These findings may be consistent with the process of "trained immunity," in which infection or vaccination yields epigenetic changes to the innate immune system. These changes, in turn, manifest as long-lived, nonspecific innate immune memory, and can lead to faster and more effective responses to subsequent infections. An experimental study in mice confirmed that infection with influenza was able to induce trained immunity, leading to protection against *S. pneumoniae* one month later[33]. In humans, trained immunity can persist for 3 months to over a year[34] Whether influenza specifically induces trained immunity in humans remains to be seen,

but the duration of the interaction effect found in this work suggests that this is an interesting question for future studies to explore.

In addition to the expected interaction effect of influenza on RSV, our results suggest that infection with RSV may lead to a similar reduction in susceptibility to influenza, albeit with a shorter duration. This was not expected based on existing research: while ferrets and mice infected with RSV experienced less severe illness due to influenza, they were not less likely to acquire influenza infection[12,15]. Additionally, a modeling study in humans suggested a trend toward an increased risk of influenza infection in the week after RSV infection, although this study made several strong assumptions concerning the duration of the interaction and the extent of susceptibility to influenza and RSV, and the result was not statistically significant[35]. These discrepancies may be partially explained by the observed circulation patterns of both viruses: since RSV outbreaks peaked after influenza outbreaks in almost all seasons in both Hong Kong and Canada, it is possible that the strength and duration of the effect of RSV on influenza is difficult to infer from these data. However, it is also worth noting that much of the work on this interaction to date has focused on the effect that infection with influenza has on RSV, and not on the effect of RSV infection on influenza. Our results indicate a need for more consideration of this direction as well.

Despite our expectation that the strength and duration of the individual-level interaction between two pathogens should be consistent over time and space, we find differences in the inferred strength of interaction between locations, particularly for the effect of influenza on RSV. It is possible that this discrepancy is due to differences in some unmodeled characteristics between locations, such as age structure, behavior of infected individuals, or circulation of other interacting pathogens. It is also worth noting that Canada is a large geographic region, which could have obfuscated the population-level signal of the interaction. Finally, it may be that, due to the complexity of interactions and their sometimes subtle impact at the population level, consistent estimates of an interaction's strength and duration are difficult to obtain when fitting to individual locations. For this reason, methods capable of pooling information from multiple locations are likely to be critical in future work. Such methods may include mixed-effects models, which allow for the estimation of parameters at several levels (e.g., location-specific vs. season-specific), and are attractive for their ability to greatly reduce the number of parameters estimated through the use of random effects[36]. While tools to apply these methods to fitting exercises using transmission models are currently in development (see, for example, panelPomp[36]), these tools are at the frontier of the field, and their application to complex modeling problems remains a challenge.

We are aware of only one other study attempting to characterize the interaction between influenza and RSV by explicitly modeling the transmission of both pathogens. Waterlow et al.[23] fit a model to influenza and RSV hospitalization data from Vietnam, and found that the data were equally consistent with primary infection leading to a 41% reduction in susceptibility to the other virus persisting for 10 days post-infection, and with no effect on susceptibility. The former finding is consistent with our estimate of interaction strength but not duration in Canada, and is only qualitatively consistent with our results in Hong Kong. These differences may be explained in part by the different assumptions and components of these two models. For example, in Waterlow's model, it was assumed that all individuals were either fully or partially susceptible to RSV at the beginning of every season, consistent with studies reporting a duration of RSV immunity of less than one year[37,38]. In contrast, we chose to estimate the proportion immune to RSV at the beginning of every season, finding strong statistical evidence against the hypothesis of full susceptibility in our data (Supplementary Table 5 and Supplementary Fig. 4). Although immunity conferred by RSV infection is imperfect, our estimates are consistent with those of a previous modeling study, which estimated a slow rate

of waning immunity after RSV infection (5–15% per year in individuals aged ≥5 years)[39]. Unlike Waterlow et al., we also accounted for seasonality in the transmission of both influenza and RSV, through the use of environmental variables (temperature and humidity) in Hong Kong and sinusoidal forcing in Canada. As seasonality is a potentially important confounder, not accounting for it may lead to biased estimates of the interaction effect. Other key model differences include Waterlow et al.'s use of an age-structured model and the assumption of a symmetric interaction. The apparent difficulty in estimating a single value for the strength of the interaction effect in Vietnam may also be due to the data used for inference: for Vietnam in particular, there were very few observed cases of influenza and RSV during the study period, and influenza outbreaks lacked clear seasonality. However, the inconsistencies in results may simply emphasize the need for methods capable of accounting for data from several locations, as estimates from distinct individual locations may differ.

We also report evidence that both temperature and absolute humidity play an important role in driving the transmission of influenza and RSV. Past findings indicate that influenza transmission increases monotonically with decreasing temperature[40–42], and that there is a U-shaped relationship between transmission and absolute humidity, such that transmission increases when absolute humidity is either low or very high[40,41]. Meanwhile, RSV transmission appears to increase with lower absolute humidity and higher precipitation[43]. Given that temperature and absolute humidity are highly correlated, it makes sense that we find a reduction in the transmissibility of influenza and RSV as temperatures increase, and an increase in the transmissibility of both viruses with increasing absolute humidity when controlling for temperature. The high values of absolute humidity observed year-round in Hong Kong may also explain the positive inferred effect of humidity. Overall, our findings suggest that weather conditions are more favorable to influenza and RSV transmission in the summer than in the winter, although the magnitude of forcing is small (Supplementary Fig. 8). This is consistent with existing work on influenza in Hong Kong[44]. Additionally, our results contribute to the existing evidence that climatic factors modulating transmission of RSV, which have not been extensively studied, are similar to those for influenza[43].

Broadly, our estimates of the season-specific reproduction numbers of both influenza and RSV are consistent with values reported in the literature[45,46]. Empirical estimates of the proportion immune at the beginning of each season are less common. However, our estimates for influenza are mostly in line with findings from past modeling studies, although for some seasons, we find relatively high levels of protection[47,48]. While in conflict with several studies suggesting that immunity to RSV is short-lived (i.e., less than a year)[37,38], our finding of moderate to high population-level immunity to RSV is consistent with studies estimating a longer duration of RSV immunity[39] and with high seroprevalence among most age groups[49,50], and may in part reflect the existence of partial immunity among those with previous exposure to RSV[51,52]. We note that our estimates of the proportion immune varied extensively between seasons. This may be because, unlike the shared parameters, these parameters can only draw on a single season's worth of information. Ideally, information on the rate of waning immunity and the previous season's attack rate would be accounted for in estimating these parameters. However, given that surveillance data represent only a small sample of actual infections, true attack rates are impossible to calculate. Furthermore, estimates of the duration of influenza and RSV immunity vary widely[37–39,53,54], and, for influenza, are highly dependent on variable rates of antigenic evolution[55,56]. Fit values of the proportion immune at the beginning of each season implicitly account for factors such as the degree of immunity remaining from previous seasons and the introduction of antigenically distinct viral strains. Encouragingly, we find that fit values of the proportion immune to influenza tend to be higher after larger influenza outbreaks,

suggesting that the fit parameters are indeed accounting for remaining immunity from the previous seasons (Supplementary Fig. 4).

In the presence of a negative interaction between influenza and RSV, vaccination with LAIV, which may induce similar protective effects as natural influenza infection[31], may lead to conflicting effects on RSV burden. We find that substantial increases in RSV attack rates may be possible, particularly in Hong Kong, perhaps due to the lower underlying RSV burden relative to influenza. However, we emphasize that the overall impact of LAIV on RSV attack rates is highly dependent on the relative timing and intensity of influenza and RSV outbreaks in a given population and season. While RSV outbreaks are more likely to precede influenza outbreaks in temperate locations[57], outbreak patterns vary widely by season in both temperate and subtropical locations. For this reason, it is unlikely that the effect of LAIV use on RSV burden will be consistent over time in any given location. Rather, public health practitioners should continuously reevaluate whether widespread LAIV use is likely to increase the RSV burden as each season progresses, and prepare accordingly.

We emphasize that, although LAIV has been shown to have a protective effect against RSV in mice[31], no studies to date have assessed the strength and duration of its impact on RSV susceptibility in humans. Given this level of uncertainty, the exact numerical impact of LAIV on RSV burden in different scenarios, as well as the exact vaccine coverage levels and timings expected to yield the lowest RSV attack rates, should be viewed with caution. Instead, the overall patterns between underlying outbreak dynamics and LAIV impact we have highlighted above should be viewed as the key results of this analysis. Future work determining the actual impact of LAIV on RSV risk at the individual level will be needed to produce more specific predictions.

The findings of our simulation study are consistent with previous work on influenza vaccines: during the 2009 influenza pandemic, observational studies in Canada reported an increase in the odds of infection with H1N1pdm09 among those vaccinated against seasonal influenza[9], while a randomized controlled trial of influenza vaccine in Hong Kong found higher rates of non-influenza infections among vaccinated children[10]. Both observations were suggested to have been driven by a lack of temporary, nonspecific immunity among those who were vaccinated, resulting from lower levels of natural infection with seasonal influenza. More generally, our results pave the way for future work on the indirect effects of vaccines, a topic that has received increasing attention throughout the COVID-19 pandemic[58,59].

Broadly, the results of this simulation study also highlight the importance of accounting for interactions between pathogens when conducting research on infectious diseases. In addition to influenza and RSV, interactions of public health relevance likely exist between influenza viruses and *S. pneumoniae*[7,8], influenza and rhinovirus[3,60], and influenza and SARS-CoV-2[61], among many others. As we have demonstrated, failure to account for these interaction effects could lead to considerable over- or underestimates of the effects of proposed public health strategies. In more extreme scenarios, the implementation of ineffective or even harmful interventions could be detrimental not only to public health, but also to public perception of infectious disease epidemiology and of modeling, fields which already suffer from a lack of trust from the public[62-65]. Where polymicrobial systems made up of interacting pathogens exist, studying individual pathogens in isolation will not be sufficient to guide informed public health planning.

Of course, in order to effectively account for interactions in public health practice, researchers and practitioners alike will require a better understanding of these interactions. Our work demonstrates the utility of mathematical models in achieving this understanding: although estimates of interaction strength differed by location, we provide evidence that influenza does indeed inhibit infection with RSV, and vice versa. We were also able to substantially narrow down the plausible ranges of values for both the strength and duration of this

interaction. Since the influence of interactions can be difficult or even impossible to discern using standard observational epidemiologic methods[6,20], it is particularly encouraging to find that transmission models can be successful where other methods fail. Critically, transmission models explicitly and mechanistically account for all components of an interaction, something that purely statistical methods are not capable of achieving. Of course, these methods will not always be effective, and identifiability issues may persist depending on the data used. As we have noted, interactions may be difficult to accurately and precisely characterize using data from only a single location. In particular, challenges may arise in locations where overlap between circulating pathogens is low, or where one pathogen consistently peaks before the other. In addition to modeling approaches, synthetic data generated from our empirically validated model can be used to assess new statistical methods for inferring interaction characteristics, an approach similar to that used in ref. 6.

Despite our promising findings, several limitations of this study are worth noting. First, our model does not include age structure, and instead assumes homogeneous mixing throughout the population. While influenza transmission is primarily driven by school-aged children[66], RSV attack rates are highest among infants and young children[28]. Depending on the extent of contact rates between these age groups, a homogeneously mixed model may be expected to either over- or underestimate the extent to which those infected with influenza come into contact with those infected with RSV, and vice versa. To assess whether the assumption of homogeneous mixing is likely to be an issue, we generated synthetic epidemic data from an age-structured model (Supplementary Fig. 17), and fit our model to these data. We found that the homogeneous mixing model was able to correctly infer the values of the interaction parameters (Supplementary Table 6), perhaps because the synthetic epidemics were synchronous across age groups (Supplementary Fig. 18). This suggests that, at least for the population and pathogens considered here, homogeneous models are sufficient to capture the broad characteristics of this interaction, even when the underlying transmission dynamics vary by age group (see Supplementary Text). This is encouraging, as the large number of unknown parameters (including initial conditions) in age-structured models makes their estimation challenging.

A second limitation is that we do not consider the potential impact of the interaction effect on disease severity, choosing to focus solely on the effect on susceptibility. Past research has yielded mixed results concerning the severity of coinfections with influenza and RSV: some studies have found coinfections to be more severe than infections with either virus alone[18,19], while others have found evidence of reduced severity[12,15]. In the modeling study of influenza and RSV cocirculation in Vietnam discussed above, the fitted model was most consistent with a 2–20 times increase in reporting among coinfected individuals[23]. We attempted to fit our model allowing for a change in infection severity among coinfected individuals, but found that this effect was not identifiable. This suggests that we have reached the limit of what our particular combination of data and model are able to infer about the interaction between influenza and RSV, but it is important that future work continue to explore this effect. Encouragingly, including an effect on severity in our model did not alter the MLEs of the other shared model parameters, suggesting that our results regarding the interaction effect on susceptibility were not compromised by not including an effect on severity.

As a final limitation, this work was highly computationally intensive. For this reason, we opted for a relatively simple model best suited to reproducing unimodal epidemics, which precluded the inclusion of influenza A(H3N2) data from Hong Kong in this study. While we would expect the effect of A(H3N2) on RSV to be similar to that of other influenza (sub)types, future work will assess this directly by developing a more complicated model of multi-peak seasonal epidemics. To

assess whether ignoring H3N2 circulation was likely to influence inferred values of the interaction parameters in Hong Kong, we conducted a sensitivity analysis in which we fit the model to only the final two seasons of data, in which H3N2 circulation was minimal. Inferred values of the interaction parameters were similar to those obtained in the main analysis (Supplementary Fig. 20). More generally, a model able to account for multi-peak epidemics would also allow for the study of interactions with additional pathogens typically associated with multiple peaks or year-round transmission. For example, rhinovirus tends to peak in both spring and autumn[67], and may reduce replication and onward transmission of influenza[60,68]. While results of a sensitivity analysis in which rhinovirus incidence was allowed to modulate the transmission of influenza are consistent with this hypothesis (see Supplementary Text), explicitly modeling rhinovirus transmission, as we have done here with RSV, would provide much more informative results concerning the strength and duration of this effect. Given the extent of overlap between influenza and rhinovirus (Supplementary Fig. 19), failure to account for rhinovirus circulation could have also biased our results, although we note that the sensitivity analysis described above did not yield substantial differences in the estimated values of the shared model parameters (Supplementary Fig. 20). For these reasons, extending the model presented here to consider alternative pairs, or a greater number, of respiratory viruses is an exciting area for future research.

In this work, we produce one of the first estimates of the strength and duration of the interaction between influenza viruses and RSV, and of the relevance of this interaction in modulating disease burden. Specifically, our results suggest that infection with either influenza virus or RSV causes a moderate to strong reduction (39 to 100%) in susceptibility to the other virus for 1 to 5 months, and that this individual-level effect can lead to substantial reductions in the attack rates of both viruses at the population level. We further explore the implications of this finding for widespread LAIV use, and find that the effect of vaccination with LAIV at the population level is highly dependent on the underlying transmission dynamics of both viruses in a given location and season. Overall, our results demonstrate that accounting for pathogen–pathogen interactions is critical, both when conducting epidemiologic research and when formulating practical strategies to reduce the burden of infectious diseases. Furthermore, we show that mathematical models are a powerful tool for improving our understanding of how these interactions operate at the individual- and population-levels.

## Methods
### Respiratory infection data
**Hong Kong.** Weekly data on influenza and RSV circulation in Hong Kong from 2014 through 2019 were obtained from the Centre for Health Protection of the Government of the Hong Kong Special Administrative Region[69]. These data consisted of the number of samples testing positive for influenza and for RSV, as well as the total number of samples tested (Fig. 1a and Supplementary Fig. 1a). Influenza virus detections were broken down by type (A or B) and by subtype (for influenza A viruses). Samples were obtained primarily from public hospitals, although some were also taken from out-patient clinics[70,71]. Typically, samples were taken from those patients for whom a specific diagnosis would be useful in informing treatment. All samples were tested for both influenza and RSV, and all testing from February 10, 2014, was done using molecular testing rather than viral culture[41].

We also obtained data on the weekly proportion of all consultations that were due to influenza-like illness (ILI), defined as fever with cough or sore throat[70], within a sentinel surveillance network of public out-patient clinics from the same source[72] (Fig. 1c, d).

We then separated the data into seasons, such that seasons began in week 46 (mid-November) and continued through week 45 of the

following year. This cutoff was chosen as earlier cutoffs often occurred mid-RSV epidemic, while later cutoffs were prone to occurring during influenza epidemics. We were, therefore, left with six seasons of data (2013–14 through 2018–19) for both viruses in Hong Kong. Although data were not available for 2013, and therefore for the first seven weeks of the 2013–14 season, we retained this season in our dataset to maximize the number of seasons available for analysis. Rather than modeling each influenza (sub)type separately, we summed weekly cases of influenzas A(H1N1) and B. Because epidemics of influenza A(H3N2) in Hong Kong often display multiple peaks per year, we did not consider influenza A(H3N2) here. This allowed us to avoid explicitly modeling loss of immunity to influenza over time, which would needlessly complicate the model and potentially compromise the fitting process.

**Canada.** Weekly data on the number of cases of and tests for influenza and RSV in Canada from 2010 through 2014 were obtained from FluWatch, the national influenza surveillance system run by the Public Health Agency of Canada (PHAC)[73] (Fig. 1b and Supplementary Fig. 1b). As in Hong Kong, influenza viruses were categorized by type and subtype, and the majority of tests were conducted on samples taken from emergency departments and hospital inpatients, with some taken from higher-risk outpatients[73]. Again, samples were typically taken when a diagnosis could potentially inform treatment. Unlike in Hong Kong, not all samples were tested for both viruses. Additionally, we obtained data on the weekly population-weighted proportion of consultations with primary care providers that were due to ILI (Fig. 1d), defined as fever and cough plus sore throat, muscle or joint pain, or fatigue[73], also from FluWatch.

In Canada, we chose a seasonal cutoff at week 35 (early September), consistent with the cutoff used by FluWatch. This left us with four seasons of data (2010–11 through 2013–14) in Canada. Again, instead of modeling each influenza (sub)type separately, we summed across all influenza A and B viruses to obtain the total weekly number of influenza cases detected.

### Transmission model
We represent the transmission of influenza and RSV throughout the model population using a deterministic, compartmental SITRxSITR model, in which compartments indicate the infection status of individuals according to both viruses (e.g., $X_{SI}$ represents the number of people susceptible to influenza and infected with RSV). This results in a total of sixteen compartments (Fig. 2). The interaction effect is modeled by reducing the force of infection of one virus among those currently (I) or recently (T) infected with the other virus (i.e., the interaction operates by reducing susceptibility to infection). Both the strength (i.e., the extent of this reduction) and duration (i.e., the amount of time spent in compartment T) of the interaction are allowed to vary according to whether an individual is initially infected with influenza or with RSV (i.e., the interaction is allowed to be asymmetric). This model is comparable to the one initially developed by ref. 20.

Because we model each yearly epidemic separately, we assume no within-season waning immunity. This assumption is in line with estimates of the duration of immunity for both influenza and RSV. Immunity to influenza is consistently estimated to last for multiple years[74,75]. While estimates for RSV are less certain, past modeling work has suggested that immunity persists for at least one year[39,53]. Furthermore, our assumption is consistent with other RSV modeling work, which also does not account for the within-season waning of immunity[23,76].

Average infectious periods were fixed at 5 days for influenza[77] and 10 days for RSV[37,78]. The size of the model population is taken to be the population size of Hong Kong or Canada during each epidemic, based on data from the Census and Statistics Department of the Government of the Hong Kong Special Administrative Region[79] and Statistics Canada[80].

**Climate forcing.** Evidence suggests that climatic conditions act as important modulators of influenza and RSV transmission. Specifically, a U-shaped effect of absolute humidity on influenza transmission has been suggested, with higher transmission occurring when humidity is either low or very high[40,41]. The effect of temperature, meanwhile, is monotonic, such that increased transmission occurs at lower temperatures[40-42]. Although RSV has received less attention, existing work suggests that temperature and humidity modulate RSV transmission in a similar manner[43,81].

In Hong Kong, we allow the force of infection for each virus to vary according to an exponential function of weekly average mean temperature and absolute humidity, such that:

$$\beta_i(t) = \hat{\beta}_i e^{\eta_{AH_i} AH(t) + \eta_{temp_i} T(t)} \qquad (1)$$

where $\eta_{temp1}$ and $\eta_{AH1}$ represent the extent to which temperature and absolute humidity, respectively, impact the transmissibility of influenza, and $\eta_{temp2}$ and $\eta_{AH2}$ represent the extent to which temperature and absolute humidity, respectively, impact the transmissibility of RSV; and $\hat{\beta}_1$ and $\hat{\beta}_2$ represent the force of infection of influenza and RSV in the absence of climate forcing. Although we do not expect the effect of absolute humidity to be monotonic based on past studies[40,41], we choose to model the relationship with a monotonic, exponential function for simplicity. We believe this simplification makes sense because (1) absolute humidity in Hong Kong is high year-round (Supplementary Fig. 3b), likely rendering the effect of low absolute humidity less important, and (2) temperate and absolute humidity are highly correlated (Supplementary Fig. 3c), such that we may expect transmissibility to increase monotonically with absolute humidity once the temperature is controlled for.

Daily mean temperature and relative humidity data were obtained from the US National Centers for Environmental Information's Global Surface Summary of the Day (GSOD) data using the R package GSODR[82,83], and absolute humidity was calculated using the Clausius–Clapeyron relation[84]. Weekly values were taken to be the mean values of temperature and absolute humidity each week. Finally, data were standardized to have a mean of zero and a variance of one. This was done both to convert the climate data into a dimensionless form, and to facilitate interpretation of the climate forcing parameters.

Because temperature and absolute humidity are highly correlated (Pearson's $r = 0.944$ over the course of the study period; see Supplementary Fig. 3), we performed a sensitivity analysis to check whether including both variables improved model fit over including temperature alone. We also compared our model to one fit using a sinusoidal forcing term, rather than climate data (see below and Supplementary Table 5).

**Sinusoidal forcing.** Because Canada covers a large geographic area over which climatic conditions vary considerably, it is unlikely that climatic conditions averaged over the country would meaningfully modulate the force of infection of influenza and RSV. For this reason, when fitting our model to data from Canada, we instead allowed the force of infection for each virus to vary according to a sinusoidal wave, such that:

$$\beta_i(t) = \beta_i \left(1 + b_i \cos\left(\frac{2\pi}{52.25}(t - \varphi_i)\right)\right) \qquad (2)$$

where $b_i$ represents the extent to which the strength of forcing varies over the year, $\varphi_i$ represents the week during which the force of infection is maximal, and $\hat{\beta}_i$ represents the average force of infection; these parameter values are allowed to differ for influenza ($i = 1$) and RSV ($i = 2$). The division by 52.25 inside the cosine function both accounts for the fact that our data are weekly, and specifies the period (in weeks) of the sinusoidal wave.

**Observation model.** Due to the prevalence of mild and asymptomatic infections, many cases of influenza and RSV are never reported, and therefore do not show up in empirical datasets. To model the process by which individuals seek healthcare and are tested for specific viral infections, we draw from a binomial distribution at each time point, where the number of trials is equal to the total observed number of tests for respiratory viruses conducted that week, and the probabilities of a positive test for influenza ($P_1(t)$) or RSV ($P_2(t)$) at time $t$ are:

$$P_i(t) = \min\left(1.0, \hat{\rho}_i(t) \frac{H_i(t)}{ILI(t)}\right) \qquad (3)$$

where:

$$\rho_i(t) = \rho_i \left(1.0 + \alpha \cos\left(\frac{2\pi}{52.25}(t - \varphi)\right)\right) \qquad (4)$$

Here, $H_i(t)$ represents the modeled incidence of virus $i$ at time $t$, while $ILI(t)$ represents the observed proportion of ILI per consultation at time $t$. Notably, while the model output $H_i(t)$ is in terms of incidence per total population, the data, $ILI(t)$ are instead in terms of incidence per all-cause consultation. Thus, these terms are not directly comparable. The parameter $\hat{\rho}_i$ is a composite parameter accounting for both the rate at which individuals infected with virus i report their illness and are diagnosed with ILI, and for the scaling by Bayes' Rule necessary to directly compare $H_i(t)$ and $ILI(t)$[47] (see Supplementary Text for more details)[47,48,85]. We account for possible seasonality in reporting and in background consultation rates using a cosine function, such that the exact value of the composite reporting and scaling parameter varies over time, with its value at time $t$ represented by $\rho_i(t)$. Thus, $1 + \alpha$ represents the maximum value of $\rho_i(t)$, while $\varphi$ represents the week during which $\rho_i(t)$ is maximal. We divide the argument of the cosine function by 52.25 to account for the fact that our data are weekly, and to impose a period of one year on reporting seasonality. To ensure that the probability of a positive test does not exceed one, we take the minimum of one and of $\rho_i(t) \frac{H_i(t)}{ILI(t)}$ at each time point.

## Model fitting

For model fitting, parameters are designated as either shared or season-specific (Table 1). Shared parameters (e.g., those describing interaction characteristics or climate forcing) are assumed not to vary by season, and are therefore constrained to take the same value in all seasons; season-specific parameters (e.g., the initial reproductive number or the proportion immune at the start of the season) are allowed to differ by season. By fitting the proportion of infected and immune at the beginning of each individual season, we avoid the need to explicitly account for processes such as the loss of immunity over time or complicated strain dynamics, which can in particular complicate the model fitting process for influenza. Meanwhile, constraining the shared parameters to take the same value in all seasons maximizes the amount of information available for the estimation of these parameters, which include our key parameters of interest. Models were fit to data from each location separately.

All model fitting was accomplished using a maximum likelihood approach to conduct trajectory matching. More specifically, model fitting was performed using a two-step process. In the first step, we obtained reasonable initial estimates for the season-specific parameters by fitting each season of data separately, assuming no interaction, climate forcing, or reporting seasonality. In the second step, we fit the model to all seasons of data for a given location simultaneously, with initial values of the season-specific parameters drawn from the ranges fit in the previous step. By constraining this second step to begin in a relatively good region of the parameter space, we allow for more efficient convergence to the MLE. To ensure convergence to the

MLE, multiple rounds of fitting to all seasons simultaneously were performed.

We obtained 95% confidence intervals for all parameter values using parametric bootstrapping[24]. This method is computationally efficient, and has been shown to perform well for similar work in the past[8,61,86]. Specifically, we generated 500 sets of synthetic epidemics at the MLE for each location using the stochastic observation model, where each set contained a simulated epidemic for all available seasons, and fit the model to each set of synthetic epidemics. Confidence intervals were obtained by computing the highest posterior density[87] of the best-fit estimates over all 500 synthetic epidemics. More specific details on the process used for model fitting and construction of the confidence intervals can be found in the Supplementary Text.

We assessed the quality of model fits by comparing the observed data to data simulated from the model at the MLE. We further confirmed convergence to the MLE by calculating the profile likelihood of $\theta_{\lambda I}$[25] (Supplementary Fig. 10).

### Simulation study of vaccine impact

In order to assess the potential for vaccination with a live attenuated influenza vaccine (LAIV) to reduce the burden of RSV, we conducted a simulation study. We adapted our SITRxSITR model, described above, to allow for vaccination of individuals who were fully susceptible to influenza, RSV, or both (specifically, those in compartments $X_{SS}$, $X_{SI}$, $X_{ST}$, $X_{SR}$, or $X_{RS}$) with LAIV (Supplementary Equation (2) and Supplementary Fig. 13). To account for the different outbreak dynamics observed in different regions, we ran our simulation study for two scenarios: a "subtropical" scenario, where all model parameters were set to their MLEs obtained when fitting the model to data from Hong Kong, and a "temperate" scenario, where parameters were set to their MLEs based on the data from Canada (Table 1). For consistency, we set the interaction parameters to the values obtained from Hong Kong in both scenarios; results from a sensitivity analysis instead using the values obtained from Canada can be found as Supplementary Fig. 15. Vaccination was assumed to confer either strong ($\theta_{\lambda vacc} = \theta_{\lambda I}$ as inferred based on the Hong Kong data) or moderate ($\theta_{\lambda vacc} = \theta_{\lambda I}$ as inferred based on the Canada data) protection against RSV; in both cases, protection against RSV due to LAIV was assumed to wane at the same rate as protection due to natural infection, again using the value obtained in Hong Kong. Additionally, vaccination conferred imperfect ("leaky") immunity to influenza[32] with a vaccine efficacy of 80%[88]. For simplicity, we assume that all vaccinated individuals receive the vaccine at a single instantaneous time point.

For each scenario, we simulated the total number of influenza and RSV cases over the course of each season at a range of vaccine coverage levels and timings using our deterministic model. We calculated the influence of LAIV on RSV burden for each simulation as a rate ratio:

$$RR = \frac{AR^1_{RSV}}{AR^0_{RSV}} \tag{5}$$

where $AR^1_{RSV}$ represents the RSV attack rate throughout the season in a partially vaccinated population, and $AR^0_{RSV}$ represents the RSV attack rate in a population where no vaccination occurred. Thus, values less than one indicate that LAIV reduced the overall attack rate of RSV for the season, while values greater than one indicate that LAIV led to an increase in the attack rate of RSV. Several sensitivity analyses, where vaccine efficacy and duration were varied, were also conducted (see Supplementary Fig. 16).

### Implementation

All analyses were conducted in R version 4.2.3[89]. The model was coded and run using the package pomp[90] (version 5.4), and fit using the sbplx algorithm via the package nloptr[91,92] (version 2.0.3). All code can be found on GitHub[93], as well as on Edmond, the Open Research Data Repository of the Max Planck Society[94].

### Reporting summary

Further information on research design is available in the Nature Portfolio Reporting Summary linked to this article.

### Data availability

The raw data used in this study are freely available online; access and use are not subject to any permissions or conditions. Specifically, the data from Hong Kong can be found at: https://www.chp.gov.hk/en/statistics/data/10/641/642/2274.html and https://www.chp.gov.hk/en/static/24015.html The data from Canada can be found at: https://search.open.canada.ca/opendata/?od-search-portal=Open%20Data&search_text=fluwatch. All code necessary to clean and process these data are published online, and can also be accessed and used without permission (see Code availability statement). Climate data from the US National Centers for Environmental Information can be downloaded using the R package GSODR (see Code availability statement). Synthetic data used in the simulation study of LAIV impact, as well as synthetic age-structured data, are available as Source Data. Source data are provided with this paper.

### Code availability

All R code used in this study can be found at: https://github.com/sarahckramer/resp_virus_interactions.

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

## Acknowledgements

We thank Laura Barrero Guevara for their helpful comments on the manuscript. S.C.K., S.P., and M.d.d.C. were funded by the Max Planck Society through the core funding received by MDC's research group. No other specific funding was received for this study.

## Author contributions

S.C.K. conducted the analyses and wrote the first draft of the initial and revised manuscripts. S.P. assisted with analyses for the revised manuscript. J.-S.C. provided content expertise. M.d.d.C. conceived of the study design and oversaw the analysis. All authors helped draft and approved the final version of the manuscript.

## Funding

## Competing interests

M.D.d.C. received postdoctoral funding (2017–2019) from Pfizer and consulting fees from GSK. The remaining authors declare no competing interests.
