## [Peer Review File · Nature Communications]

Reviewers' Comments:

Reviewer #1:

Remarks to the Author:

Kramer et al present an interesting modelling study assessing interactions between influenza and RSV infections over 5 seasons in Hong Kong. The extensive discussion particularly focusses on potential impact of non-specific cross protection, the authors highlighting a remarkable reduction in RSV susceptibility following influenza A H1N1 infection that persists for 3–4 months and a similar impact for the converse effect which lasts around 2–3 months.

The manuscript is well written with a detailed introduction and discussion and well thought-through modelling.

Comments:

1. As the authors note, this work is not the first to model these interactions. Waterlow et al PLoS Comput Biol (June 2022) reported mathematical modelling based on 11 years of Vietnamese data, concluding that there was a short lived (10 day) period of cross-protection. However, they concluded that absence of either pathogen was not to the detriment of the other, and that vaccination against either pathogen is unlikely to have a major effect on the burden of disease caused by the other. The important differences between this and the current study are described on lines 313-333. The current study would be strengthened and of broader applicability if they were able to access other datasets and validate predictions in other contexts.
2. The discussion is very extensive but well written, running to almost 10 pages. Throughout the manuscript there is some scope to merge some sentences and sections to improve readability (lines 62–65, for example). If the Editors were to ask for a reduction in word count, where would the authors be prepared to compromise?
3. While aspects of the models are mostly explained well for a non-specialist audience a concise summary (or visual abstract) of the dominant parameters that drive predictions (in addition to the provided table) would enhance immediate understanding for many readers.
4. Comparisons of predictions of model with observed values over the season and seasons should be included. Currently, only a comparison of predicted versus observed cases is included. Even when only considering total case numbers the models seem to show poor performance with a wide range in predicting RSV case numbers. The authors comment that this may be due to the weaker seasonality of RSV, which raises questions over the accuracy of results presented and whether the approaches used are suitable for modelling epidemics that do not have simple large and seasonal peaks. Again, application of the model developed by the authors to other parts of the globe would be of great interest.
5. The authors excluded modelling of interactions with H3N2 on account of its multi-peak complexity. How many cases of H3N2 occurred during the study period and (assuming these would also provide non-specific protection) how major might be the effect of this omission? The authors themselves point this out as a limitation of (in their own words) "simple models", can more complex models be used to assess this?
6. The model estimates concerning the different effects of humidity and temperature on influenza A and B seems questionable: is this more likely a shortcoming of the models and data than a real effect?
7. Why are the ranges of immunity estimates so wide (line 188)? Again, is this a case of pushing the model beyond what the data can accurately inform. The range of 0% to 71% for RSV when fit alongside flu B is particularly notable, though other ranges are similarly wide, making it hard to interpret.

8. In the Discussion, the authors refer to several viral and bacterial pathogens that have significant interactions with influenza, i.e., rhinovirus and pneumococcus. How frequent were these infections during the study period, do the authors have access to such data, and might omission of other common infections in the model terms not undermine the accuracy of modelling results?

9. The authors exclude seasons with under 2000 influenza cases, with the justification that virus-virus interaction would be minimal. How was this 2000 case figure chosen, how does it compare to the average number of cases and, most importantly, why can fitting not be performed for these below 2000 case seasons to provide evidence of minimal interaction during times of lower influenza?

10. Although not mentioned in the title, the authors give prominence to the potential impact of LAIV in some parts of the manuscript. While this is an interesting and potentially important point and a nice use of the model, there appears to be no actual data on LAIV to support the argument. The authors have instead simply applied a proportion of the live infection effect to model LAIV. There are several unknowns surrounding this approach. We suggest that the LAIV section and the resulting conclusions (especially those in the Abstract) should be caveated and scaled back.

Reviewer #2:

Remarks to the Author:

The manuscript "Characterizing the epidemiological interactions between influenza and respiratory syncytial viruses and their implications for epidemic control" describes an attempt to infer interaction between flu and RSV from population level syndromic respiratory surveillance in Hong Kong using mathematical modelling. With the arrival of a number of potential RSV prevention strategies and a drive to enhance influenza vaccination coverage in the wake of the SARS-CoV-2 pandemic a better understanding on such potential interaction could prove very important to anticipate ecological knock-on effects.

While the topic is important, I do have a number of concerns that in sum make it difficult for me to recommend this work for publication. The main ones are

- There seems not a great deal of appreciation that the type of data available for this work is really not ideal to make inference on the causal relationship of flu and RSV infections and that even with the best of models one needs to be cautious in its interpretation. As a result, much of the discussion and conclusions seems over-confident as to how much this study adds to the overall evidence base – in particular since this work proposes a strength of interaction that by far exceeds any of the previous estimates be it from time series, mouse models or others.
- The data analysed is partly selected post-hoc. Only seasons with >2000 flu cases are included which is arbitrary and should be avoided but more importantly H3 is excluded because its seasonal pattern could not be easily modelled. However, H3 is particularly interesting as its circulation seems to overlap much with that of RSV making the hypothesis of competitive exclusion difficult unless one proposes that H3 would behave completely different from other flu strains
- The interaction of RSV with either influenza B or H1 are considered in isolation. Flu B and H1 in the considered setting and time frame share relatively similar seasonality and thus it is little surprise that both models come to similar conclusions: both flu B and H1 exert strong competitive pressure on RSV and with this observed RSV seasonality can be well replicated in either model. The logical flaw in this is that if indeed H3 does compete strongly with RSV and substantially reduces RSV transmission, then this will need to be accounted for in the fluB-RSV model because otherwise the effect will be double counted.

I do have several additional concerns, questions and comments that I have listed in no particular order below:

- General:

o This modelling study uses data from Hong Kong. However there is no mention of any input from Hong Kong on the interpretation of the data. Collaborating with local experts often helps limiting the scope for misinterpretation and in this case should have also allowed for potentially useful information, eg information on how many samples were infected with both flu and RSV at the same time and the age of the cases. I very much encourage the authors to consider this approach in the future but realise that for the current work it is too late.

o It is a very difficult project to infer individual level effects on infection from population level observations on disease. So, one needs to be careful and generally sceptical in its conduct and interpretation which is the basic approach for my review.

- Introduction:

o Several studies have also suggested that RSV-flu coinfection enhances severity! Need to better balance the reporting of the evidence here.

o Both Pfizer and GSK have announced that in phase 3 trials their candidates met the primary endpoints and thus while true that there is currently no vaccine licensed this will likely change in the imminent future making it highly unlikely that LAIV will be trialled for RSV prevention - so the authors should temper respective statement on the potential importance of LAIV for RSV prevention given that those gains from LAIV are largely hypothetical.

o I wouldn't call mentioned modelling study inconclusive; it was able to exclude a large part of the potential parameter space and found support in the data for two potential regions of interest.

- Results

o The clear seasonality for flu is not true for H3 so you will need to clarify that you are only talking about H1 and B and that this is not true for H3.

o Before you throw everything into the model it would be interesting to see some more straight forward data analyses. Eg your modelling results would suggest that flu and RSV incidence are negatively correlated. This may be masked by seasonality but worth a look what the data says directly

o "the effect of RSV on influenza B appears to be unidentifiable" I would re-phrase this as "no evidence for ..."

o It is hardly a surprise that the model fit will be slightly better when an additional parameter is fitted. So, you will need to strengthen your argument if you want to include both temp and humidity in the main model despite their strong correlation, at least a DIC would be more meaningful than a straight up comparison of goodness of fit

- Methods

o Why remove seasons with fewer than 2,000 flu cases? This seems arbitrary and also an unnecessary loss of available data. I would think this data should be included in the inference. Or at the very least I would need to see some sensitivity analysis on this.

o Similarly, one needs to be careful about excluding H3N2 because of the shape of the epidemics. Unless there is a clear biological reason for why interaction with or the epidemiology of H3N2 would be different this would seem to be epidemiological cherry picking; particularly so since H3 constitutes the majority of all flu cases and thus, unless a very different effect on RSV it should have a large impact on RSV and the other inferences. Also H3 seems to circulate much more in the RSV season than the other flu variants.

o The data description should be improved. E.g. it is unclear to the reader what would trigger a test for RSV and for flu in the outpatient clinics? E.g. is there mandatory testing for all ILI cases in the country, is this sentinel surveillance, is the tested proportion of ILI cases dependent on the season, are all samples tested for both flu and RSV, if not what determines this, ...?

o This seasonal forcing function that translates mean temperature and absolute humidity into changes in the FOI would be good to visualise for at least the fitted nu parameters. Also, assuming that flu and RSV are driven by exactly the same weather determinants is not unreasonable but quite a strong assumption. In the worst case the model will need to trade off this seasonal forcing by artificially (?) using other parameters (like competition) to reproduce the observed difference in seasonality. Thus It would be good to look at the sensitivity of this assumption eg by using a sinusoidal curve allowing for differences in phase for the two pathogens

o It is not clear whether the cosine seasonal reporting rate is needed for HongKong; ideally this should be clarified with local expertise on the the way surveillance is conducted

o In eqn 2 do you really mean $\cos(\cos())$ or is this a typo? Also in Eqn 2 is this meant to be 1.0 times pi or something else? Actually, there are a couple of things here that I don't understand. Are $\pi(t)$ and π - are they the same? In the para below you refer to phi in another way to the phi in the equation. The 52.25 is unmotivated (scaling to weekly I guess).

o

o Why would the reproduction number vary by season? I agree that the net R will vary due to different immunity but that is captured separately. I may be missing something here but I would think that the R_0 should be estimated as a joined parameter across seasons

o Is the susceptibility to infection at the start of the season estimated independently from the attack rate of the previous season? I could see why this may be necessary for the fitting to be tractable but it should be highlighted as a limitation.

o Why 99% CI's? That's untypical. Please provide a rationale or stick to typical reporting of 95%

o The fitting process is odd. Please elaborate why no more usual MCMC inference approach is used. Also, I am pretty sceptical that this method is indeed making sure that not just a local maximum is captured given the complexity of a 47 dimensional parameter space. More information is needed on the fitting - ie how many fitting attempts did you do with how many and how different starting values and did they indeed all come to the same conclusion? Also I would like to see a sensitivity analysis where the authors fix the interaction effect to 1, ie no effect, and then run the otherwise same inference pipeline. Is the model still able to fit and how much worse is the MLE? I see what you have done with the profile LL and Figure S8, but I think a sense check of no interaction and the rest of the parameters fitted would be interesting.

o the model does not have age groups. You do some discussion around that but need to add that AR for flu is typically highest in School age whereas for RSV in v young kids and infants - hence the overlap may be less than included in this model.

o The covid pandemic has shown that RSV seasonality is quite easily disrupted. This in turn implies that any observed seasonality is to a great extent driven by the build up and loss of immunity. It should be discussed how the absence of this in this model could influence results.

o I am struggling to understand the approach to incorporating the data. The authors provide a number of equations, but these are convoluted, and I am unsure whether they indeed add up to something meaningful. Particularly I am not seeing any plain English justification to why the data on proportion of consultations that are flu/rsv is a helpful quantity. The model will use infection incidence and a typical approach is that modelled infection incidence should be the same as observed ILI incidence times $P(\text{reported ILI} | \text{infection})$. The latter may be seasonal, similar to what you seem to estimate with the cosine function. Given that we have no clue what else was causing GP consultations during that time period I am struggling to understand what the proportion of consultations that we diagnosed as ILI would add to our understanding to the flu/rsv epidemiology and hence why to include this in the fitting. Please expand on that.

o The proportion immunity against reinfection at the beginning of a RSV is suspiciously high given that most estimates find immunity against reinfection to only last for $<1y$; ie most should be susceptible again the next year.

- Discussion

o The discussion of the Waterloo paper needs some balancing. I don't think it is entirely clear which of the two approaches is the more robust one. Waterloo had info on co-infection and estimated separately the interaction of infection and of severity which is ignored in this work - would be good to discuss this a little. I don't think it is fair to call the assumption that everyone was susceptible at the beginning of the RSV season "arbitrary" - I think only citing 37 as a counter argument is not a fair reflection of all available evidence. See my earlier comments re the use of temperature and humidity as seasonality predictors - I think this argument for bias can really go either way. Finally, the reason why Waterloo assumed symmetric of interaction was that RSV always preceded flu and hence the interaction in the other direction could not be independently assessed. Long story short: please provide a more balanced discussion of relative strength and weaknesses

o The authors suggest that more data from other settings is needed to make more robust inference. The data from the Waterloo paper is published. It would be interesting to see if the different models indeed come to different conclusions from the same data or whether the

difference is in the data. But this is probably for another paper.

o As $Re = R_0 \cdot (1 - \text{immune})$, how is it possible in this model to estimate both simultaneously from syndromic surveillance data? Can you show the correlation plots for these as well please? Also, I don't quite understand why R_0 should vary between seasons - can you please elaborate on this?

o I think the authors will need to temper their conclusion on the RSV influenza interaction - this study is one piece of evidence and I do not believe that it would necessarily be so robust as to disregard conflicting evidence.

Reviewer #3:

Remarks to the Author:

This is an interesting, well conceived, well written, and compelling article that examines time series of epidemiological observations for evidence of interaction between influenza and respiratory syncytial virus (RSV) at the scale of a whole population. This is a timely topic and the implications of the paper are potentially very important. In particular, the authors adduce evidence for immunological interaction among these circulating pathogens with appreciable effects at the population scale. This in turn suggests that the use of live attenuated influenza vaccines may be effective in suppressing RSV, though this will depend on the recent past history of prevalence in both pathogens. The paper explains these dependencies in some detail and thereby offers some initial guidance toward the design of possible vaccination strategies and supporting surveillance efforts. It is therefore likely to be of broad interest in the vaccinology and public health communities, as well as for the broader public.

Although the evidence for the existence of these effects, as presented, is compelling, there are nevertheless improvements that can be made both to clarify the presentation and to solidify the conclusions. I detail these in the following.

I 88: Please say more about the "inconclusive results" of the earlier study based on data from Vietnam. The reader naturally wonders at this point what these results were. Of course, it would distract from the presentation to say too much here, but some indication of what was not concluded in the earlier study is called for. See also below.

II 104ff: It is concerning that the authors exclude A(H3N2) data from consideration. The justification given—that the models cannot handle multiple peaks per season—exacerbates rather than assuages this concern. I miss some discussion of what aspects of the modeling approach put the consideration of multiple peaks beyond its ken.

I 106: Why is 2000 cases chosen as the threshold for including a season? What would the consequences of altering this threshold be?

I 124: Why do the authors choose to use a deterministic model? What would the consequences of including stochasticity be? To be clear, I think it would be unreasonable to re-analyze the data using a stochastic model. However, the authors have sufficient experience with stochastic models that they should be able to speak to the question of what the consequences of the assumption of deterministic dynamics are likely to be. Perhaps this is best addressed in the Discussion section.

II 141–151: The authors point out several parameters that are unidentifiable given the data they have. It would be interesting to have some idea as to why the data are silent about these parameters when they are not with respect to others. For example, can the authors say something about features absent in the data that would help to identify these parameters were they present? Conversely, can we begin to understand which features in the data contribute to the identification of those parameters that are identifiable?

II 188–193: How is the observed evidence in favor of variability in start-of-season population

immunity to be interpreted? Is this evidence for long-lived immunity coupled with antigenic evolution, as we have by now come to expect? If not, is it inconsistent with this? If so, would it be possible and perhaps better to model this immunity dynamically instead of via extra parameters? More generally, why do the authors choose to introduce start-of-season parameters instead of the more usual dynamical approach in which all parameters of the model are either constants and initial conditions?

ll 322–324: Similarly, in this passage, the choice to make the fraction of the population susceptible at the beginning of the season is presented as an alternative to Waterlow’s decision to make this a constant. Of course, it is an alternative, and a more reasonable one, but it is not the only alternative, nor is it the most intuitive one. Why do the authors take this approach?

One way to address this would be to discuss the observed differences in season-specific parameters from one season to the next (Supp Fig 3) and to plot these against estimates of the total number of within-season infections. In other words, how does the estimate of S at the beginning of a season depend on the total number of infections in the preceding season?

ll 195ff & Fig 3: The R^2 values quoted amount to a comparison of the model’s predictions relative to those of a model in which the cases are constant through time. It would be informative to similarly present a comparison with a more descriptive model, such as one which captures the average seasonality.

Supp Fig 6: The profile likelihoods could be solidified. More exploration is clearly needed at very low values of these parameters and it may be necessary, for example, to use a logarithmic horizontal scale. Also, a proper profile likelihood plot should have a vertical scale showing at most 10 or 20 log units—at least if the Wilks theorem applies—since one log unit of resolution is then meaningful. Finally, the variability in the estimates for the Influenza B-RSV interaction cast doubt on the identification of the MLE.

Minor remarks

l 48: I believe “host cell” should be hyphenated here.

l 50: It would be better not to begin two successive paragraphs with the same word.

l 76ff: As written, this sentence is awkward. I suggest recasting as “It is, as usual, unclear whether and how results from studies in animal models are applicable to human populations. Moreover, seemingly intuitive study designs, ...”

l 78: What is meant by “phase differences”? Is this meant to describe differences in the timing of epidemic peaks between different infections? As it stands, this is unclear.

l 81: It is unclear what is meant by the “large number of components” needed to characterize interactions. Please clarify.

ll 82–83: Mathematical models are useful not only because they can express complex and nonlinear dynamics and account for various mechanisms, but because they establish logical linkages between these things and observable data.

l 137: The point estimate is reported to 3 significant figures, but the uncertainty does not support this. Perhaps a formulation like “...reduction in susceptibility persists for roughly 100 days (point estimate 111 d, 95% CI 96–127 d)...” would be better. There are other instances of similarly overprecise statements.

l 148: Given that the duration of potential effect of RSV on Influenza B is not identifiable, it is not surprising that the magnitude is similarly unidentified. Indeed, it would be strange if it were

otherwise, no?

ll 165, 171: “[C]ontribute significantly” is not the right language in this context. It would be better to say that temperature and absolute humidity both “modulate transmission”. Similarly, in l 171, temperature and humidity cannot be spoken of as “drivers” of transmission, but rather as “modulators”.

l 180–181: The mathematical symbols have not yet been defined and use unfamiliar notation. In particular, notation such as R_{20} is annoyingly idiosyncratic. It would be preferable to write R_{20} .

Make sure all mathematical symbols are defined in the text as they are first used (e.g., l 148, ll 180–181).

l 285–286: It would be better to say “...we found evidence for an interaction effect...”

ll 351–352: “...we expect that all influenza (sub)types will be influenced by climate to some extent.” Indeed, this is consistent with your findings, as would be worth reminding us here.

ll 428–429: “...a previous modeling study found that the impact of influenza on susceptibility to RSV could not be inferred using data from Vietnam.” The reader naturally wonders why the earlier study failed. Some explanation, or even speculation, would be appreciated here.

ll 446–448: “This suggests that homogeneous models are sufficient to capture the broad characteristics of this interaction, even when the underlying transmission dynamics vary by age group...” Unless I am mistaken, this will not be true in general. Indeed, if the coupling between age groups is weak, there may be substantial differences between the predictions of an age-structured and a homogeneous model. As written, this conclusion appears more general than I believe the authors may intend.

l 596: There appears to be a typo in Eqn 2.

REVIEWER COMMENTS

We would like to thank all of the reviewers for their detailed and helpful comments. Before responding to each reviewer's individual comments, we want to highlight the major changes we have made to the manuscript:

- 1. We now fit the model to data from Canada, in addition to data from Hong Kong, in order to test whether our results were consistent in various locations. We found that these data appear to be consistent with a weaker but similarly long-lasting interaction effect than the data in Hong Kong. These results are now incorporated into our main text; furthermore, we discuss possible reasons for the differing results in both locations, as well as potential future analyses for identifying a consensus (lines 314-326). Parameter estimates for both locations can be found in Table 1, which is copied and pasted below this list for convenience.*
- 2. Because of the observed challenges when fitting the model to data on influenza B and RSV, including the fact that our main outcome of interest was not identifiable using only these data, we have elected to combine influenza A(H1N1) and influenza B in Hong Kong, and to run our fitting procedure on this combined influenza data and RSV, rather than fitting individual influenza subtypes separately. Broadly, the new results are consistent with our original results, and many parameters are more precisely fit, likely due to the greater amount of data we consider at once.*
- 3. In light of the recent approval of vaccines against RSV for older adults and pregnant people, we have reframed the section of our manuscript on the use of LAIV to focus on the potential tradeoffs of widespread LAIV use, rather than on LAIV as a potential RSV control measure. Specifically, we explore, broadly, which scenarios are more likely to lead to an increase in RSV burden, and which are more likely to lead to a decrease. Additionally, we emphasize that, in the absence of concrete information concerning the impact of LAIV on RSV susceptibility in humans, this analysis is meant to highlight general trends on the expected impact of LAIV use, and not to predict the exact quantitative impact in any specific scenario.*
- 4. We have conducted several additional sensitivity analyses to test the robustness of the model fits to several assumptions we have made. Specifically, we tested a sinusoidally-forced model in Hong Kong, and attempted to account for the circulation of influenza H3N2 and of rhinovirus. Encouragingly, we find that the estimated values of the shared parameters obtained in these sensitivity analyses were similar to those reported in the main text. Furthermore, we assessed the importance of including the interaction effect and the extent of existing immunity to RSV in our model by fitting models with no interaction effect and where the model population was fully susceptible to RSV at the beginning of each season (as in Waterlow et al. 2022). Both of these models were found to fit the data significantly worse than our main model.*

Table 1. Descriptions and maximum likelihood estimates of all model parameters.

Parameter	Description	Fit Value (95% CI)		Season-Specific?
		Hong Kong	Canada	
$\theta_{\lambda 1}$	Strength of the interaction effect of influenza on RSV	1.2×10^{-9} (0, 3.2×10^{-4})	0.61 (0.51, 0.62)	No
$\theta_{\lambda 2}$	Strength of the interaction effect of RSV on influenza	6.2×10^{-6} (0, 0.13)	0 (0, 0.013)	No
δ_1	Rate of loss of cross-protection against RSV after influenza infection	0.065 (0.063, 0.073)	0.092 (0.049, 0.22)	No
d_2	Rate of loss of cross-protection against influenza after RSV infection, relative to δ_1	3.0 (2.3, 3.0)	3.9 (1.3, 5.5)	No
ρ_1	Composite reporting rate/scaling parameter for influenza	0.20 (0.18, 0.20)	2.7 (2.7, 3.2)	No
ρ_2	Composite reporting rate/scaling parameter for RSV	0.034 (0.032, 0.035)	0.84 (0.79, 0.92)	No
η_{temp1}	Impact of temperature on influenza	-0.15 (-0.15, -0.13)	NA	No
η_{AH1}	Impact of absolute humidity on influenza	0.23 (0.21, 0.23)	NA	No
η_{temp2}	Impact of temperature on RSV	-0.089 (-0.12, -0.083)	NA	No
η_{AH2}	Impact of absolute humidity on RSV	0.19 (0.18, 0.22)	NA	No
b_1	Amplitude of seasonal forcing on influenza	NA	0.19 (0.18, 0.19)	No
b_2	Amplitude of seasonal forcing on RSV	NA	0.15 (0.14, 0.16)	No
φ_1	Week of maximum seasonal forcing on influenza	NA	26.2 (25.7, 26.5)	No
φ_2	Week of maximum seasonal forcing on RSV	NA	30.2 (29.0, 30.5)	No
α	Amplitude of seasonality in reporting and background consultation rates	0.61 (0.57, 0.64)	0.90 (0.90, 0.91)	No
φ	Week of maximum ρ_1/ρ_2	43.8 (43.3, 44.3)	48.0 (47.9, 48.2)	No
Ri_1	Initial effective reproductive number for influenza	See Supplementary Fig. 4		Yes
Ri_2	Initial effective reproductive number for RSV	See Supplementary Fig. 4		Yes
I_{10}	Initial proportion of the population infected with influenza	See Supplementary Fig. 4		Yes
I_{20}	Initial proportion of the population infected with RSV	See Supplementary Fig. 4		Yes
R_{10}	Initial proportion of the population immune to influenza only	See Supplementary Fig. 4		Yes
R_{20}	Initial proportion of the population immune to RSV only	See Supplementary Fig. 4		Yes
R_{120}	Initial proportion of the population immune to both influenza and RSV	See Supplementary Fig. 4		Yes

Reviewer #1 (Remarks to the Author):

Kramer et al present an interesting modelling study assessing interactions between influenza and RSV infections over 5 seasons in Hong Kong. The extensive discussion particularly focusses on potential impact of non-specific cross protection, the authors highlighting a remarkable reduction in RSV susceptibility following influenza A H1N1 infection that persists for 3–4 months and a similar impact for the converse effect which lasts around 2–3 months.

The manuscript is well written with a detailed introduction and discussion and well thought-through modelling.

Thank you for your supportive comments.

Comments:

1. As the authors note, this work is not the first to model these interactions. Waterlow et al PLoS Comput Biol (June 2022) reported mathematical modelling based on 11 years of Vietnamese data, concluding that there was a short lived (10 day) period of cross-protection. However, they concluded that absence of either pathogen was not to the detriment of the other, and that vaccination against either pathogen is unlikely to have a major effect on the burden of disease caused by the other. The important differences between this and the current study are described on lines 313-333. The current study would be strengthened and of broader applicability if they were able to access other datasets and validate predictions in other contexts.

We agree that our work would be significantly strengthened by applying our fitting procedure to data from a variety of locations. Applying our model as-is to the data from Vietnam specifically is not possible, as our model makes use of syndromic data (e.g., ILI data), and it is unclear whether these data are available from Vietnam (see also our response to Reviewer 2's second point regarding our Discussion). However, as mentioned above, we were able to find publicly-available data on influenza and RSV circulation, including ILI data, from Canada, and include the results of fitting our model to these data in the revised manuscript. These data were consistent with a weaker interaction than the data in Hong Kong, but with a similar duration (see point 1 in the numbered list at the beginning of this document).

2. The discussion is very extensive but well written, running to almost 10 pages. Throughout the manuscript there is some scope to merge some sentences and sections to improve readability (lines 62–65, for example). If the Editors were to ask for a reduction in word count, where would the authors be prepared to compromise?

Thank you for pointing out lines 62-65, which were unintentionally redundant. We have condensed these sentences, and have gone through the manuscript and attempted to condense in other locations as well.

We have tried to be as comprehensive as possible for the initial submission, but understand that the discussion might need to be condensed. In this case, we would continue to discuss the key results on the strength and duration of interaction, as well as the potential impacts of LAIV, in detail, while truncating the discussion of additional results, such as the role of climate forcing and the inferred season-specific parameters. Our discussion of the broader significance of this work as it relates to the study of pathogen-pathogen interactions and the use of mathematical models to characterize interactions (lines 419-445) could also be reduced.

We also note that, because the interaction parameters were not always identifiable when we modeled subtypes of influenza separately, we have elected to combine influenza A(H1N1) and B in the revised manuscript. This has led to many sections of the manuscript already being shortened.

3. While aspects of the models are mostly explained well for a non-specialist audience a concise summary (or visual abstract) of the dominant parameters that drive predictions (in addition to the provided table) would enhance immediate understanding for many readers.

We have added two panels to Fig. 2 (Fig. 2b-c), in which we show how varying values of the interaction strength and duration parameters influence epidemic trajectories. We hope that this will help readers to obtain a more intuitive feel for the meaning of these key model parameters.

4. Comparisons of predictions of model with observed values over the season and seasons should be included. Currently, only a comparison of predicted versus observed cases is included. Even when only considering total case numbers the models seem to show poor performance with a wide range in predicting RSV case numbers. The authors comment that this may be due to the weaker seasonality of RSV, which raises questions over the accuracy of results presented and whether the approaches used are suitable for modelling epidemics that do not have simple large and seasonal peaks. Again, application of the model developed by the authors to other parts of the globe would be of great interest.

In order to hopefully partially address your concern, we now compare basic epidemic metrics, specifically peak timing, peak intensity, and observed attack rates, between simulated and observed outbreaks. Specifically, we generated 100 simulations of each seasonal outbreak at the MLE, then calculated the percentage of simulations that reproduced the observed peak timing within 2 weeks, and the percentage of simulations that reproduced peak intensity and attack rate within 25%. Results can be found in Supplementary Table 4. Specifically, we found that total attack rates in particular are captured with high accuracy in both locations. Peak intensity is also frequently captured accurately, although accuracy suffers in seasons with weaker signal. Simulations of influenza are typically also accurate at capturing the timing of the observed peak; in many seasons, 100% of simulations correctly replicated the observed peak timing). RSV simulations, on the other hand, often fail to capture the true peak timing, particularly in Hong Kong. However, this metric is not particularly meaningful for outbreaks where case counts remain fairly consistent throughout the season, as with most RSV outbreaks in our Hong Kong dataset. Thus, although the model cannot always closely reproduce the observed cases from week to week, the model at the MLE does accurately reproduce overall epidemic patterns. This is encouraging, as capturing the general transmission patterns of both viruses is likely more important than whether the model can closely predict the exact weekly case counts for all weeks.

Finally, as you say, application of the model to other locations is important. Regarding this, please see the first point in the numbered list at the beginning of the response letter. In particular, outbreaks of RSV in Canada show much clearer seasonality than RSV outbreaks in Hong Kong, and the resulting model fit to the RSV data in Canada is therefore much higher (Fig. 3d). The inconsistent results between locations concerning the strength of the interaction effect of influenza on RSV could, like you say, be due to the weaker seasonality of RSV in Hong Kong. Indeed, we now note that it could be difficult to obtain consistent estimates of interaction characteristics by fitting individual locations in isolation, and emphasize the importance of fitting to data from several regions (lines 318-326).

5. The authors excluded modelling of interactions with H3N2 on account of its multi-peak complexity. How many cases of H3N2 occurred during the study period and (assuming these would also provide non-specific protection) how major might be the effect of this omission? The authors

themselves point this out as a limitation of (in their own words) “simple models”, can more complex models be used to assess this?

The number of H3N2 cases over the study period in Hong Kong is shown in Supplementary Fig. 2. Notably, the amount of H3N2 circulation varied widely from season to season. To determine whether ignoring H3N2 was likely to substantially alter our results, we refit our model for only the two seasons where minimal circulation of H3N2 occurred, 2017-18 and 2018-19. We found that the inferred values for the strength and duration of the interaction were similar for the two analyses, although the inferred duration of the effect of RSV on influenza wr

6. The model estimates concerning the different effects of humidity and temperature on influenza A and B seems questionable: is this more likely a shortcoming of the models and data than a real effect?

Yes, this seems more likely to indicate that these particular data do not allow these parameters to be accurately inferred, rather than that the effect of climate on influenza varies substantially by subtype. There could also be difficulty in estimating these parameters due to the high correlation between temperature and humidity. However, as we noted, the majority of the past experimental work we found focusing on climate forcing of influenza has been conducted on influenza A, not influenza B. It is possible that further research would indeed reveal that the climate drivers of influenza B are different.

As mentioned in the numbered list at the beginning of this document, we now combine data on influenza A(H1N1) and influenza B in Hong Kong, rather than fitting the two subtypes separately. This helps to increase our chances of precisely estimating our parameters of interest, as there is more data being incorporated and more signal in the data season-to-season. In this case, the results for climate forcing are more consistent with the original A(H1N1)-RSV results. As further support for our findings concerning the seasonal pattern of climate forcing (Supplementary Fig. 8), we found a very similar pattern of forcing when fitting a model where transmission was forced using a sinusoidal wave, rather than using climate data explicitly.

7. Why are the ranges of immunity estimates so wide (line 188)? Again, is this a case of pushing the model beyond what the data can accurately inform. The range of 0% to 71% for RSV when fit alongside flu B is particularly notable, though other ranges are similarly wide, making it hard to interpret.

Because the season-specific parameters are being fit on the basis of individual seasons, it is likely that they are less precisely and accurately fit than the shared parameters, which are able to take information from multiple seasons of data simultaneously. Since our main goal was to estimate the interaction parameters, we elected to keep the additional features of the model relatively simple to avoid having to fit an unreasonably large number of parameters, but this decision of course comes with tradeoffs. Given the simplicity of the model, it is possible that the fit values of the reproductive numbers and the proportion of the population immune are accounting for other features of a population that may help drive outbreak dynamics that are not explicitly modeled here. For example, the proportion immune may account for widespread partial immunity, or year-to-year differences in contact patterns in the population. While it is therefore encouraging that the ranges of fit values are broadly realistic, the exact fit values of these parameters may not be as reliable as those of the parameters that are shared between seasons, which are able to take much more data into account when being estimated. We have now included more discussion of the difficulty of inferring these season-specific parameters (lines 376-386).

8. In the Discussion, the authors refer to several viral and bacterial pathogens that have significant interactions with influenza, i.e., rhinovirus and pneumococcus. How frequent were these infections during the study period, do the authors have access to such data, and might omission of other common infections in the model terms not undermine the accuracy of modelling results?

Our dataset included information on rhinovirus incidence. While there is overlap of influenza and rhinovirus circulation, influenza outbreaks tended to peak between outbreaks of rhinovirus in both Hong Kong and Canada (see new Supplementary Fig. 19). Because of the extent of overlap in the circulation of these pathogens, it is possible that ignoring rhinovirus could bias our results, a point that we now make in our limitations (lines 490-491).

That said, we have now run a sensitivity analysis in Hong Kong where we allow rhinovirus incidence to modulate the force of infection of influenza. Specifically, the force of infection of influenza was multiplied by $e^{-\beta_{rhino} inc_{rhino}}$, where inc_{rhino} represents the extent of rhinovirus transmission at time t , defined as the proportion of tests positive for rhinovirus multiplied by the ILI rate per consultation at time t , standardized to have a mean of 0 and variance of 1, and β_{rhino} is a parameter determining the extent to which circulating rhinovirus restricts influenza transmissibility. We found that the maximum likelihood estimates of the shared parameters obtained using this model are very similar to those values obtained in the main text (see Supplementary Fig. 20). We now mention this sensitivity analysis in lines 486-493.

Although it is likely that an interaction exists between influenza and pneumococcus, the evidence suggests that the interaction is unidirectional. In other words, although influenza may increase the rate of pneumococcal acquisition and progression to invasive disease (Domenech de Cellès et al. (2019), PNAS), there is no evidence for an impact of pneumococcus on influenza transmission or severity. For this reason, we do not expect ignoring pneumococcus incidence to bias our results.

9. The authors exclude seasons with under 2000 influenza cases, with the justification that virus-virus interaction would be minimal. How was this 2000 case figure chosen, how does it compare to the average number of cases and, most importantly, why can fitting not be performed for these below 2000 case seasons to provide evidence of minimal interaction during times of lower influenza?

The other reviewers had the same feedback. We have now removed this restriction, and its mention in the paper, and re-run the model to provide updated parameter estimates.

10. Although not mentioned in the title, the authors give prominence to the potential impact of LAIV in some parts of the manuscript. While this is an interesting and potentially important point and a nice use of the model, there appears to be no actual data on LAIV to support the argument. The authors have instead simply applied a proportion of the live infection effect to model LAIV. There are several unknowns surrounding this approach. We suggest that the LAIV section and the resulting conclusions (especially those in the Abstract) should be caveated and scaled back.

We have now emphasized the lack of empirical data in our discussion of these results, and have noted that the reader should focus on the qualitative, rather than the exact quantitative, results from this analysis (lines 401-408; see also point 3 in our numbered list of major revisions at the beginning of the response letter). We have also edited the wording in the abstract to emphasize that these results are based on an assumption about the effect of LAIV on susceptibility to RSV.

Reviewer #2 (Remarks to the Author):

The manuscript “Characterizing the epidemiological interactions between influenza and respiratory syncytial viruses and their implications for epidemic control” describes an attempt to infer interaction between flu and RSV from population level syndromic respiratory surveillance in Hong Kong using mathematical modelling. With the arrival of a number of potential RSV prevention strategies and a drive to enhance influenza vaccination coverage in the wake of the SARS-CoV-2 pandemic a better understanding on such potential interaction could prove very important to anticipate ecological knock-on effects.

While the topic is important, I do have a number of concerns that in sum make it difficult for me to recommend this work for publication. The main ones are

- There seems not a great deal of appreciation that the type of data available for this work is really not ideal to make inference on the causal relationship of flu and RSV infections and that even with the best of models one needs to be cautious in its interpretation. As a result, much of the discussion and conclusions seems over-confident as to how much this study adds to the overall evidence base – in particular since this work proposes a strength of interaction that by far exceeds any of the previous estimates be it from time series, mouse models or others.

We have attempted to mitigate the inherent limitations of surveillance data by incorporating both virologic and syndromic data, a decision we now justify in lines 106-109. We also now mention a further limitation of surveillance data in lines 378-380. In general, we have attempted to include more caveats throughout, and to emphasize that additional work will be needed moving forward to confirm our results.

That said, we emphasize that fitting transmission models to surveillance data is a common method of inferring key epidemiologic parameters, and that similar methods have been used successfully to better understand pathogen-pathogen interactions in the past (see Shrestha et al. (2013) and Domenech de Cellès (2019) on interactions between influenza and Streptococcus pneumoniae, Domenech de Cellès et al. (2021) on the interaction between influenza and SARS-CoV-2, Noori and Rohani (2019) on the interaction between measles and pertussis, and, of course, Waterlow et al. (2022) on the interaction between influenza and RSV).

Our result suggesting that influenza can almost completely inhibit RSV is consistent with experimental results in both ferrets (Chan et al. (2018)) and mice (Drori et al. (2020)), both of which found complete or almost complete prevention of RSV infection shortly after infection with influenza, although our estimated duration is longer than the duration found in either of these studies (as we note in lines 284-289). However, it is unclear to what extent we expect the duration of this interaction to be consistent across species. With regard to estimates from time series, these are extremely prone to failure, as we discuss in lines 77-79 and 125-126.

As a test of the robustness of our findings, we now also include results obtained from fitting our model to data from an additional location, Canada, where we find a more moderate impact of influenza on RSV (specifically, a reduction in RSV susceptibility by about 40%). While this estimate differs quantitatively with our estimate from Hong Kong, it suggests that our conclusion regarding the existence of a negative interaction between influenza and RSV is robust. Furthermore, our estimates of the duration of this effect, as well as the strength and duration of the impact of RSV on influenza, are similar in both locations (see Table 1). Finally, we discuss potential reasons for the quantitative differences in estimates between locations, as well as the need for future work to help identify a consensus (lines 314-326).

- The data analysed is partly selected post-hoc. Only seasons with >2000 flu cases are included which is arbitrary and should be avoided but more importantly H3 is excluded because its seasonal pattern could not be easily modelled. However, H3 is particularly interesting as its circulation seems to overlap much with that of RSV making the hypothesis of competitive exclusion difficult unless one proposes that H3 would behave completely different from other flu strains

This comment was raised by the other reviewers as well and, as addressed above, we have removed the >2000 case restriction. Additionally, we have added a sensitivity analysis to assess whether the exclusion of H3N2 may impact our results; specifically, we fit the model for only those seasons where H3N2 circulation is low. More details can be found in response to point 5 of Reviewer 1's comments, copied and pasted here:

"To determine whether ignoring H3N2 was likely to substantially alter our results, we refit our model for only the two seasons where minimal circulation of H3N2 occurred, 2017-18 and 2018-19. We found that the inferred values for the strength and duration of the interaction were similar for the two analyses, although the inferred duration of the effect of RSV on influenza was longer than in the main analysis."

Finally, we want to emphasize that the significant overlap in the circulation of RSV and H3N2 does not in any way suggest a lack of a negative interaction (see lines 77-79 and 125-126).

- The interaction of RSV with either influenza B or H1 are considered in isolation. Flu B and H1 in the considered setting and time frame share relatively similar seasonality and thus it is little surprise that both models come to similar conclusions: both flu B and H1 exert strong competitive pressure on RSV and with this observed RSV seasonality can be well replicated in either model. The logical flaw in this is that if indeed H3 does compete strongly with RSV and substantially reduces RSV transmission, then this will need to be accounted for in the fluB-RSV model because otherwise the effect will be double counted.

It is true that, in choosing to model individual influenza subtypes separately, we are ignoring the impact of the unmodeled subtypes. Because we also found that our main parameters of interest were difficult to infer using the data on influenza B in particular, we have revised the main analysis to consider the sum of the influenza A(H1N1) and B data in Hong Kong, rather than analyzing the two separately (see point 2 in our numbered list at the beginning of this document).

I do have several additional concerns, questions and comments that I have listed in no particular order below:

- General:

- o This modelling study uses data from Hong Kong. However there is no mention of any input from Hong Kong on the interpretation of the data. Collaborating with local experts often helps limiting the scope for misinterpretation and in this case should have also allowed for potentially useful information, eg information on how many samples were infected with both flu and RSV at the same time and the age of the cases. I very much encourage the authors to consider this approach in the future but realise that for the current work it is too late.

Thank you for this comment and we agree that it is desirable to have input from local experts. The first author of this paper, S. Kramer, has previous experience modeling influenza seasonality with this Hong Kong dataset in collaboration with local experts (Yuan et al., 2021).

- o It is a very difficult project to infer individual level effects on infection from population level

observations on disease. So, one needs to be careful and generally sceptical in its conduct and interpretation which is the basic approach for my review.

We agree that one must be careful not to overinterpret this type of work. We have gone through the manuscript and attempted to soften the language so as not to come across as overconfident (see our response to your first main comment above). That said, past simulation studies, along with modeling studies fitted to real-world data, have demonstrated that such work, while difficult, can accurately and precisely infer the individual-level interaction effects between two pathogens (see references 20 and 22 for examples of simulation studies, as cited in lines 80-83; for studies on real-world data, our response to your first major comment above).

- Introduction:

- o Several studies have also suggested that RSV-flu coinfection enhances severity! Need to better balance the reporting of the evidence here.

Thank you for pointing this out. We have updated the introduction to clarify that studies have found conflicting results concerning the severity of coinfections with influenza and RSV, and to indicate that the existing evidence remains inconclusive (lines 67-69). That past studies have found conflicting evidence on the severity of coinfections is also touched upon in the discussion (lines 463-467).

- o Both Pfizer and GSK have announced that in phase 3 trials their candidates met the primary endpoints and thus while true that there is currently no vaccine licensed this will likely change in the imminent future making it highly unlikely that LAIV will be trialled for RSV prevention - so the authors should temper respective statement on the potential importance of LAIV for RSV prevention given that those gains from LAIV are largely hypothetical.

We have reframed our analysis of the potential impact of LAIV in light of the recent approval of an RSV vaccine for older adults and pregnant people, as well as RSV monoclonal antibodies for infants (see point 3 in the numbered list at the beginning of our response letter). While LAIV may still represent an interesting way of controlling RSV until vaccines are approved for more age groups, we now focus instead on the potential tradeoffs of LAIV use, given that our simulation study showed that LAIV use may actually lead to increases in RSV burden in the presence of a negative interaction between influenza and RSV. In particular, we highlight the need for both medical and public health practitioners to be aware that widespread LAIV use may have unintentional effects on RSV burden.

- o I wouldn't call mentioned modelling study inconclusive; it was able to exclude a large part of the potential parameter space and found support in the data for two potential regions of interest.

We now describe the results of the Waterlow et al. study more specifically, which also addresses a similar comment from Reviewer 3 below.

- Results

- o The clear seasonality for flu is not true for H3 so you will need to clarify that you are only talking about H1 and B and that this is not true for H3.

We have revised the manuscript to specify that we are referring to the seasonality of H1N1 and B influenzas (line 109-110).

- o Before you throw everything into the model it would be interesting to see some more straight forward data analyses. Eg your modelling results would suggest that flu and RSV incidence are negatively correlated. This may be masked by seasonality but worth a look what the data says

directly

We have updated our first results section to present some initial analysis of the correlation between influenza and RSV incidence, as well as the season-to-season difference in peak timing of the two viruses (lines 120-125). Specifically, we find that the time series are not significantly correlated in Hong Kong (Kendall's tau = -0.067; 95% CI: -0.14-0.008), and are moderately positively correlated in Canada (Kendall's tau = 0.62; 95% CI: 0.57-0.66). Influenza outbreaks almost always peaked before outbreaks of RSV, with a median difference in timing of 20 weeks in Hong Kong and 8.5 weeks in Canada (see also Supplementary Table 1).

However, we note that a negative interaction between influenza and RSV on the individual level does not necessarily imply that incidence will be negatively correlated at the population level (see Shrestha et al. (2011), PLoS Comput Biol). Indeed, the fact that nonlinear epidemic dynamics sometimes lead to unintuitive outbreak patterns at the population level is a key reason why mathematical models that account for this lack of linearity are so important in understanding the underlying drivers of transmission. This strength of mechanistic models relative to more traditional epidemiologic study designs is discussed in lines 77-89 in the Introduction. We now re-emphasize this point after presenting the results of the simple correlation between the influenza and RSV time series (lines 125-126).

o “the effect of RSV on influenza B appears to be unidentifiable” I would re-phrase this as “no evidence for ...”

As explained above, we have removed the subtype-specific analyses and now focus on the sum of influenza A(H1N1) and B.

o It is hardly a surprise that the model fit will be slightly better when an additional parameter is fitted. So, you will need to strengthen your argument if you want to include both temp and humidity in the main model despite their strong correlation, at least a DIC would be more meaningful than a straight up comparison of goodness of fit

In our comparison of the goodness of fit, we specifically performed a likelihood ratio test, which accounts for the difference in the number of parameters included in each model. This was not clear from the original text of our manuscript, and we have updated the text accordingly (line 169). Since our main analysis is not Bayesian in nature, we cannot use a DIC here, but we have added AIC values to Supplementary Table 5 to further assist in model comparison.

- Methods

o Why remove seasons with fewer than 2,000 flu cases? This seems arbitrary and also an unnecessary loss of available data. I would think this data should be included in the inference. Or at the very least I would need to see some sensitivity analysis on this.

We have removed this restriction and re-run the models.

o Similarly, one needs to be careful about excluding H3N2 because of the shape of the epidemics. Unless there is a clear biological reason for why interaction with or the epidemiology of H3N2 would be different this would seem to be epidemiological cherry picking; particularly so since H3 constitutes the majority of all flu cases and thus, unless a very different effect on RSV it should have a large impact on RSV and the other inferences. Also H3 seems to circulate much more in the RSV season than the other flu variants.

We agree that simply removing H3N2 from consideration without further sensitivity analysis is problematic. Our response to one of your previous comments discusses a new sensitivity analysis we have conducted in order to account for H3N2 circulation. Additionally, we note that we did not observe significantly more overlap in the data between H3N2 and RSV than between H1N1 + B and RSV (Kendall's tau 0.031 (95% CI: -0.044-0.11) vs. -0.067 (95% CI: -0.14-0.008)).

o The data description should be improved. E.g. it is unclear to the reader what would trigger a test for RSV and for flu in the outpatient clinics? E.g. is there mandatory testing for all ILI cases in the country, is this sentinel surveillance, is the tested proportion of ILI cases dependent on the season, are all samples tested for both flu and RSV, if not what determines this,...

Virologic surveillance in Hong Kong is conducted primarily using tests obtained from hospitals, and not from outpatient clinics. Testing was primarily performed to aid diagnosis and treatment, and not necessarily according to any more structured surveillance plan. This was consistent throughout the duration of the study period. The same samples are tested for both influenza and RSV, as well as a range of other respiratory pathogens. In Canada, testing was also primarily performed on samples from hospital inpatients and emergency departments, but not all samples were tested for both viruses. We have added this information to the Methods (lines 519-523 and 544-547).

o This seasonal forcing function that translates mean temperature and absolute humidity into changes in the FOI would be good to visualise for at least the fitted nu parameters. Also, assuming that flu and RSV are driven by exactly the same weather determinants is not unreasonable but quite a strong assumption. In the worst case the model will need to trade off this seasonal forcing by artificially (?) using other parameters (like competition) to reproduce the observed difference in seasonality. Thus it would be good to look at the sensitivity of this assumption eg by using a sinusoidal curve allowing for differences in phase for the two pathogens

The requested plot is now included as Supplementary Fig. 8. We found that the overall extent of forcing was highest in May-August and lowest in October-December, although the absolute magnitude of the forcing remained relatively low throughout the year for both viruses. This finding is consistent with that of Ali et al. (2022), who also found that climate conditions in Hong Kong were more favorable to influenza transmission in the summer than in the winter, despite Hong Kong frequently experiencing winter outbreaks. This is potentially because, unlike in many temperate locations, the temperature and absolute humidity experienced during Hong Kong's winters are rarely low enough to yield strong forcing of influenza transmission.

As suggested, we also fit a model using sinusoidal rather than climate forcing, and allowing for different amplitudes and phases for both viruses. Timing of maximum and minimum forcing were quite similar to the values obtained from the main analysis, and the phases for both viruses were also similar (36.0 for influenza and 42.2 for RSV at the MLE), suggesting that the timing of maximum transmission does not substantially vary for the two viruses. Estimates of the interaction strength parameters were similar to the main analysis for the effect of RSV on influenza, but slightly weaker ($\theta_{\lambda,1} = 0.15$) for the effect of influenza on RSV. The best-fit values for the duration of the interaction were a bit shorter for the impact of influenza on RSV (about 80 days), and longer for the impact of RSV on influenza (about 70 days) (Supplementary Fig. 20). As assessed by log-likelihood and AIC, the sinusoidal forcing model fit slightly better than the climate-forced model (Supplementary Table 5); however, when assessed by the coefficient of efficiency (as in main text Fig. 3), fit quality was very similar to the main model (overall $R^2 = 0.89$ for influenza and $R^2 = 0.59$ for RSV). Because we are interested in the impact of temperature and humidity on the transmission of influenza and RSV, we continue to present the climate-forced model as our main analysis. However, it is encouraging to see

that the results obtained using the sinusoidal forcing model are broadly consistent with the results obtained from the climate-forced model.

o It is not clear whether the cosine seasonal reporting rate is needed for HongKong; ideally this should be clarified with local expertise on the the way surveillance is conducted

It is difficult to tell whether there is, in reality, seasonality in the rate of consultations made for ILI and for all causes in Hong Kong or Canada. Theoretically, though, if this seasonality is not present, the parameter α would be 0 (or close to 0) at the MLE. By including this term, we allow the data to inform the model as to whether such seasonality is important to include or not.

o In eqn 2 do you really mean $\cos(\cos())$ or is this a typo? Also in Eqn 2 is this meant to be 1.0 times pi or something else? Actually, there are a couple of things here that I don't understand. Are $\pi(t)$ and π - are they the same? In the para below you refer to phi in another way to the phi in the equation. The 52.25 is unmotivated (scaling to weekly I guess).

Thank you for bringing this typo to our attention. We have now corrected equation 2.

Equation 2 (now Equation 3) is meant to be the minimum of 1 and the presented equation. We have clarified this in lines 655-656, and have clarified the weekly scaling (as well as the imposition of a one-year period for reporting seasonality) in lines 653-654.

We have renamed some parameters and added an equation to attempt to make the information on ρ_i and $\rho_i(t)$ clearer. Specifically, ρ_i is the baseline value of the reporting/scaling parameter, and $\rho_i(t)$ is now defined as the specific value of the reporting/scaling parameter at time t (given that there may be seasonality in reporting and all-cause consultations).

o Why would the reproduction number vary by season? I agree that the net R will vary due to different immunity but that is captured separately. I may be missing something here but I would this that the R_0 should be estimated as a joined parameter across seasons

Rather than estimating R_0 , the basic reproductive number, we have chosen to estimate the effective reproductive number at the beginning of each season. This is because this value more directly influences the transmission dynamics of a virus than does the R_0 . Furthermore, many published estimates of the reproduction number are in fact values of R effective, and not of R_0 [Biggerstaff et al. (2014), BMC Infect Dis.]. Season-specific R_0 s for our model can be calculated from the effective reproductive numbers and the proportion of the population susceptible at the start of each season.

We note that, although R_0 is often discussed as a constant quantity for a given pathogen, it is more realistic to allow for R_0 to change over time. This is because, in addition to the inherent transmissibility of a pathogen, R_0 also accounts for factors like the underlying contact patterns in a population, which are likely to change over time (for example, with school terms and holidays). Furthermore, since influenza at least is a rapidly evolving virus, it makes sense that transmissibility will vary from year to year as new strains begin to circulate.

o Is the susceptibility to infection at the start of the season estimated independently from the attack rate of the previous season? I could see why this may be necessary for the fitting to be tractable but it should be highlighted as a limitation.

We estimate susceptibility at the beginning of a season without considering the previous season's attack rate partially because it is impossible from observational data to know the true attack rate

during a given season, and because unmeasured factors other than the previous season's attack rate, such as lingering immunity from previous seasons and vaccination rates, will also influence susceptibility at the season's start. We acknowledge that, if more complete information on attack rates and existing immunity were available, it would be preferable to take such data into account when estimating the initial proportion immune. We have updated our discussion of our limitations to reflect this (lines 378-387).

o Why 99% CI's? That's untypical. Please provide a rationale or stick to typical reporting of 95%

We have updated our results to use the typical 95% confidence intervals.

o The fitting process is odd. Please elaborate why no more usual MCMC inference approach is used. Also, I am pretty sceptical that this method is indeed making sure that not just a local maximum is captured given the complexity of a 47 dimensional parameter space. More information is needed on the fitting – ie how many fitting attempts did you do with how many and how different starting values and did they indeed all come to the same conclusion? Also I would like to see a sensitivity analysis where the authors fix the interaction effect to 1, ie no effect, and then run the otherwise same inference pipeline. Is the model still able to fit and how much worse is the MLE? I see what you have done with the profile LL and Figure S8, but I think a sense check of no interaction and the rest of the parameters fitted would be interesting.

At each step of the model fitting process, we made 500 fitting attempts, each with different starting values. For the first round of fitting, where only the reporting rates and season-specific parameters were fit, start values were chosen from the broad ranges shown in Supplementary Table 2. For all subsequent rounds of fitting, start ranges were taken from the best-fit models from the previous round. For the main analysis, we performed two rounds of fitting to all seasons simultaneously, each using start values obtained from the previous round, before the MLE was reached. A final round of fitting was then run. This process is discussed in more detail in the Supplementary Text. Because of this extensive search, we feel confident that we have reached the global maximum. As further confirmation, we are able to obtain values close to the MLE when performing parametric bootstrapping; were our model-inference system prone to fitting to local maxima, this would not be the case.

In addition to the maximum likelihood method discussed in the manuscript, we also attempted to fit the model using an MCMC approach. However, perhaps because of the large number of parameters being fit, it was difficult to optimize the step sizes taken for each parameter, leading to inefficient mixing and a failure to converge.

Thank you for your suggestion regarding a sensitivity analysis where no interaction is assumed. We agree that this is a useful additional check of whether our model indeed supports the existence of the interaction we describe. We ran this analysis and found that model fit was significantly worse than when the interaction term was also allowed to be fit (AIC 24526 vs. 19471). Specific results can now be found in Supplementary Table 5. Additionally, the fit values and 95% confidence intervals for all shared parameters across a range of sensitivity analyses can now be seen in Supplementary Fig. 20.

o the model does not have age groups. You do some discussion around that but need to add that AR for flu is typically highest in School age whereas for RSV in v young kids and infants - hence the overlap may be less than included in this model.

The different attack rates and different degrees of overlap between influenza and RSV circulation in different age groups were important reasons why we wanted to test whether our homogeneously-

mixed model was capable of inferring parameters from data generated by a population with age structure. We have updated our discussion of this sensitivity analysis to better reflect this (lines 448-452).

o The covid pandemic has shown that RSV seasonality is quite easily disrupted. This in turn implies that any observed seasonality is to a great extent driven by the build up and loss of immunity. It should be discussed how the absence of this in this model could influence results.

We now discuss the fact that immunity from previous seasons is not considered in our model as part of our limitations (lines 378-380). However, we also emphasize that our estimation of the proportion of the population immune to each virus at the beginning of each season should, theoretically, account for any build up and loss of immunity occurring over previous seasons (lines 382-384). Furthermore, we now demonstrate that our fit values for the proportion immune to influenza at the beginning of a given season tend to be higher when the previous influenza season was larger, as expected (Supplementary Fig. 5). Although we do not observe the same pattern for RSV, this could be due to the fact that RSV outbreaks are more similar in size across seasons. As for within-season loss of immunity, it is common for models of individual RSV seasons to ignore this (see references 23, 39, and 46).

o I am struggling to understand the approach to incorporating the data. The authors provide a number of equations, but these are convoluted, and I am unsure whether they indeed add up to something meaningful. Particularly I am not seeing any plain English justification to why the data on proportion of consultations that are flu/rsv is a helpful quantity. The model will use infection incidence and a typical approach is that modelled infection incidence should be the same as observed ILI incidence times $P(\text{reported ILI} \mid \text{infection})$. The latter may be seasonal, similar to what you seem to estimate with the cosine function. Given that we have no clue what else was causing GP consultations during that time period I am struggling to understand what the proportion of consultations that we diagnosed as ILI would add to our understanding to the flu/rsv epidemiology and hence why to include this in the fitting. Please expand on that.

In both Hong Kong and Canada, most virologic testing is performed on samples taken from inpatients and emergency departments; in other words, the virologic data reflects only the most severe cases in the population, and may not be fully representative of respiratory virus circulation in the population as a whole. The ILI data, on the other hand, are taken from outpatient clinics and primary care physicians, and are therefore most likely to be representative of the general pattern of transmission in the population. For this reason, we believe that including both types of data yields the most complete picture of influenza and RSV circulation in these populations. We have now clarified our decision to use both virologic and ILI data in lines 106-109.

Of course, ideally we would take into account ILI per total population under surveillance. Unfortunately, only data on ILI per consultation was available, so this is what we have used. Because of this, however, our data and our model output become mismatched - the model returns cases per population, while our data are cases per consultation. To convert the model output to be in the same form as the observed data, we use Bayes' Rule, as explained in the Supplementary Text. This is what we meant when we originally described the parameters ρ_1 and ρ_2 as being part "scaling parameters," although we agree that this was not particularly clear. We have now described the purpose of these parameters in more detail in the Methods in lines 647-654. Further explanation remains in the Supplement, and can also be found in references 47 and 48, which we have now cited in the Methods.

o The proportion immunity against reinfection at the beginning of a RSV is suspiciously high given that most estimates find immunity against reinfection to only last for <1y; ie most should be susceptible again the next year.

Although it is true that many studies place the duration of immunity to RSV as being well under one year, some studies instead find a much longer duration of immunity (van Boven et al. (2020), for example, found a rate of immunity loss of 5-15% per year in most age groups), and many sources also suggest that partial immunity, at least, is widespread in the population (White et al. (2007), Henderson et al. (1979)). Our high estimates of immunity may in part reflect this partial immunity, which we do not account for explicitly. Notably, one study found that seropositivity rates in Sao Paulo, Brazil reached 100% by the age of 5 (Cox et al. (1998)). We have expanded our discussion of these results to note that, while they are in agreement with reports of extensive partial immunity to RSV, they may be in conflict with results suggesting that protection against reinfection is short-lived (lines 372-376).

- Discussion

o The discussion of the Waterloo paper needs some balancing. I don't think it is entirely clear which of the two approaches is the more robust one. Waterlow had info on co-infection and estimated separately the interaction of infection and of severity which is ignored in this work - would be good to discuss this a little. I don't think it is fair to call the assumption that everyone was susceptible at the beginning of the RSV season "arbitrary" - I think only citing 37 as a counter argument is not a fair reflection of all available evidence. See my earlier comments re the use of temperature and humidity as seasonality predictors - I think this argument for bias can really go either way. Finally, the reason why Waterlow assumed symmetric of interaction was that RSV always preceded flu and hence the interaction in the other direction could not be independently assessed. Long story short: please provide a more balanced discussion of relative strength and weaknesses

Reading through this section again, we agree that we have come across as overly critical of Waterlow et al.'s work. We have reworded the paragraph to primarily highlight differences in assumptions and methodology that may have contributed to the different results found, without necessarily naming either set of assumptions as better. Regarding the assumptions around existing RSV immunity, we include citations pointing to both short-lived and long-lived immunity to RSV, to hopefully help illustrate why different decisions were made in the two papers. In addition to differences in model assumptions, we also now point to differences in the data from the two countries as being a potential reason why the interaction effect appeared to be more difficult to estimate in Vietnam, and why parameter estimates might differ. Finally, given that our new results from Canada differ somewhat from our results in Hong Kong, we point out that it could simply be difficult to reach consistent estimates in different locations without fitting multiple locations simultaneously. Currently, tools to fit mixed effects models to multiple locations simultaneously are at the frontier of the field, and their application remains a challenge (see our commentary in lines 318-326).

The fact that we have not included an interaction effect on disease severity is discussed as a limitation of our work in lines 462-473, and we note that this is a quantity that Waterlow et al. were able to estimate. We did attempt to include this effect, but it was unfortunately not identifiable for our specific combination of data and model.

o The authors suggest that more data from other settings is needed to make more robust inference. The data from the Waterlow paper is published. It would be interesting to see if the different model s indeed come to different conclusions from the same data or whether the difference is in the data.

But this is probably for another paper.

We agree that this would be an interesting and important analysis, although it looks as though the two models also accept different types of data. In particular, our model requires both virologic data (i.e., the number of tests positive for influenza and RSV) and syndromic data (here, data on influenza-like illness rates). It is unclear whether the latter are available in Vietnam.

However, we were able to find publicly-available data on influenza and RSV positivity, as well as ILI rates, from Canada, and fit our model to these data as well. As discussed in the numbered list at the beginning of this document, as well as in response to some of your previous comments, our results concerning the existence of a negative interaction between influenza and RSV were robust, but the exact parameter values, and in particular the inferred strength of the effect of influenza on RSV, differed by location. We briefly discuss several possible reasons for these discrepancies (lines 314-320); in particular, since interactions may have a subtle impact on pathogen circulation at the population level, it may be difficult to reach a single, consistent estimate of an interaction's strength and duration unless multiple locations are fit simultaneously.

In an attempt to assess the extent to which different model assumptions (vs. differences in the data) contributed to the differences in results, we fit our model to the Hong Kong data again, this time assuming, as in Waterlow et al., that the entire model population is susceptible to RSV at the beginning of each season. We found that this model fit the data significantly less well than did our model, in which season-specific immunity to RSV was fit (Supplementary Table 5), although interaction parameter estimates did not appear greatly affected (Supplementary Fig. 20), suggesting that at least assumption alone was not primarily responsible for the differences between the two manuscripts.

o As $R_e = R_0 * (1 - \text{immune})$, how is it possible in this model to estimate both simultaneously from syndromic surveillance data? Can you show the correlation plots for these as well please? Also, I don't quite understand why R_0 should vary between seasons - can you please elaborate on this?

Although R_0 and the proportion of a population immune to a disease are indeed related, it is common for models to estimate both simultaneously. In the same way that R_0 and the proportion immune can be estimated by the same model (and frequently are), so too can R_{eff} and the proportion immune. We now include the correlation plots for the effective reproductive numbers and the proportion of the population immune to one or both viruses as Supplementary Fig. 7. Correlations between R_{i1} and $R_{10} + R_{120}$, and between R_{i2} and $R_{20} + R_{120}$, are typically relatively weak, with the exception of the strong, negative correlation between R_{i1} and $R_{10} + R_{120}$ for the 2017-18 season in Hong Kong, suggesting that there is not a strong tradeoff in values required in order to fit both quantities simultaneously.

We have discussed a few reasons why R_0 may vary between seasons above, in response to one of your comments on the Methods.

o I think the authors will need to temper their conclusion on the RSV influenza interaction - this study is one piece of evidence and I do not believe that it would necessarily be so robust as to disregard conflicting evidence.

We now point out that, because outbreaks of RSV tend to peak after outbreaks of influenza in our data, it may be difficult for our model to infer the strength and duration of the interaction effect in this direction (lines 305-307); this is further emphasized in lines 441-443. We have also mentioned an additional conflicting estimate in lines 301-304.

Reviewer #3 (Remarks to the Author):

This is an interesting, well conceived, well written, and compelling article that examines time series of epidemiological observations for evidence of interaction between influenza and respiratory syncytial virus (RSV) at the scale of a whole population. This is a timely topic and the implications of the paper are potentially very important. In particular, the authors adduce evidence for immunological interaction among these circulating pathogens with appreciable effects at the population scale. This in turn suggests that the use of live attenuated influenza vaccines may be effective in suppressing RSV, though this will depend on the recent past history of prevalence in both pathogens. The paper explains these dependencies in some detail and thereby offers some initial guidance toward the design of possible vaccination strategies and supporting surveillance efforts. It is therefore likely to be of broad interest in the vaccinology and public health communities, as well as for the broader public.

Thank you.

Although the evidence for the existence of these effects, as presented, is compelling, there are nevertheless improvements that can be made both to clarify the presentation and to solidify the conclusions. I detail these in the following.

I 88: Please say more about the “inconclusive results” of the earlier study based on data from Vietnam. The reader naturally wonders at this point what these results were. Of course, it would distract from the presentation to say too much here, but some indication of what was not concluded in the earlier study is called for. See also below.

We now briefly mention the main results of this earlier study in the introduction, rather than simply referring to the results as inconclusive. Specifically, we write that the earlier study “found that data were equally consistent with either a moderate negative interaction or no interaction” (lines 88-89).

II 104ff: It is concerning that the authors exclude A(H3N2) data from consideration. The justification given—that the models cannot handle multiple peaks per season—exacerbates rather than assuages this concern. I miss some discussion of what aspects of the modeling approach put the consideration of multiple peaks beyond its ken.

In order to test whether the exclusion of H3N2 from our analysis was likely to bias our results, we have now performed a sensitivity analysis. A description of this analysis and its findings can be found in our response to Reviewer 1’s fifth main comment, copied and pasted here:

“To determine whether ignoring H3N2 was likely to substantially alter our results, we refit our model for only the two seasons where minimal circulation of H3N2 occurred, 2017-18 and 2018-19. We found that the inferred values for the strength and duration of the interaction were similar for the two analyses, although the inferred duration of the effect of RSV on influenza was longer than in the main analysis.”

In order to capture multi-peak outbreaks of influenza, a model would likely need to account for several additional drivers of transmission, including loss of immunity and strain dynamics. Estimates for the duration of immunity to both influenza and RSV are highly variable, and, for influenza, vary by subtype (see citations in lines 380-382). Even in the case of seasonal influenza in temperate regions,

where a single outbreak occurs each year, relatively complex models are needed to adequately reproduce the size and timing of each outbreak (see Yaari et al. (2013), J R Soc Interface). Given that the exact dynamics of influenza and RSV immunity loss were not the primary outcome of interest for this study, we felt that modeling them explicitly would add needless complexity to our model. For this reason, we elected to use a simpler, single-season model, accepting that one drawback of such a model would be a relative inability to handle multi-peak outbreaks. We now justify this decision in our Methods (lines 537-538).

I 106: Why is 2000 cases chosen as the threshold for including a season? What would the consequences of altering this threshold be?

We have now removed this restriction.

I 124: Why do the authors choose to use a deterministic model? What would the consequences of including stochasticity be? To be clear, I think it would be unreasonable to re-analyze the data using a stochastic model. However, the authors have sufficient experience with stochastic models that they should be able to speak to the question of what the consequences of the assumption of deterministic dynamics are likely to be. Perhaps this is best addressed in the Discussion section.

Our decision to use a deterministic rather than a stochastic model was primarily driven by the complexity of our model fitting exercise. With several seasons of data and 54/40 parameters to fit in Hong Kong and Canada, respectively, even a deterministic approach proved to be expensive in terms of time and computational power required. For the main model in Hong Kong, for example, fitting the model using a single starting parameter set could take up to 3.7 hours (1.15 hours on average). Given that several rounds of fitting were required to reach the MLE, and that we performed a wide range of sensitivity analyses, this amounted to fitting the model thousands of times. Fitting the model in Canada, as well as for some of the sensitivity analyses, required even more time. Although this was possible using high-performance and parallel computing, each round of fitting was still quite slow. Fitting a stochastic model instead would compound the complexity of the problem, requiring substantially more time and power to fit. Furthermore, given that our analysis involves fitting both parameters shared between seasons as well as parameters allowed to vary by season, fitting a stochastic model would likely require a panel approach. Although tools for accomplishing this are being developed (see Bretó et al. (2019), J Am Stat Assoc), they are relatively new and there have been few applications to date.

It is difficult to say what exactly the consequences are of our using a deterministic rather than a stochastic model for this analysis. However, because we are modeling infection transmission in a large population, we expect the influence of stochasticity in the transmission process, at least, to be small. To illustrate this, we ran 100 simulations at the MLE for both influenza and RSV for all seasons in Hong Kong, and compared the resulting outbreaks to the average expected based on the deterministic model. In most seasons, the runs from a fully stochastic transmission model (in gray) are typically very similar to the mean deterministic trajectories (in black):

ll 141–151: The authors point out several parameters that are unidentifiable given the data they have. It would be interesting to have some idea as to why the data are silent about these parameters when they are not with respect to others. For example, can the authors say something about features absent in the data that would help to identify these parameters were they present? Conversely, can we begin to understand which features in the data contribute to the identification of those parameters that are identifiable?

Given the difficulty we had inferring values for our parameters of interest using data on influenza B and RSV, we have elected to combine data on H1N1 and B influenza in Hong Kong and fit the model to this combination of subtypes, rather than to each subtype separately (see point 2 in the numbered list at the beginning of this response letter). The main results of the manuscript have been updated accordingly. When fitting to the combined influenza data and RSV, all parameters of interest are identifiable.

However, an analysis of what features of a dataset allow for certain parameters to be precisely inferred would indeed make for an interesting analysis. Intuitively, we expect that interaction parameters would be more difficult to infer if there was little overlap in the transmission of the two pathogens of interest, or if one pathogen consistently peaked either several weeks before or after the other, ideas that we have brought up in lines 304-307 and 441-443. Indeed, we initially attempted to fit our model to data from France as well, but, as RSV outbreaks preceded influenza outbreaks in every season except the 2009 pandemic, we found that the interaction parameters were not identifiable. However, a more rigorous simulation study would be needed to definitively test these expectations, and would make for an informative future study.

ll 188–193: How is the observed evidence in favor of variability in start-of-season population immunity to be interpreted? Is this evidence for long-lived immunity coupled with antigenic evolution, as we have by now come to expect? If not, is it inconsistent with this? If so, would it be possible and perhaps better to model this immunity dynamically instead of via extra parameters? More generally, why do the authors choose to introduce start-of-season parameters instead of the more usual dynamical approach in which all parameters of the model are either constants and initial conditions?

From the fits of the start-of-season parameters alone, it is unfortunately impossible to say what factors underlie the fit values for population immunity. Certainly the existence of immunity at the beginning of each season is consistent with long-lived immunity in conjunction with antigenic evolution, but we cannot say to what extent these processes are, in reality, responsible for the values we have inferred. Most likely, the parameters for start-of-season immunity reflect a range of underlying features and processes, including lingering immunity from previous seasons, long-lived partial immunity, and the introduction of new, antigenically-distinct viral strains. This is now better explained in lines 382-387.

For this work, we have chosen start-of-season parameters specifically because they can capture this wide range of underlying drivers of immunity, without the need to explicitly model features like loss of immunity over time, the extent of partial immunity, etc. Especially for influenza, it is challenging to explicitly model changes in immunity over time - see our response to your comment about ll 104ff above. In particular, to accurately capture influenza transmission dynamics, modeling loss of immunity alone is likely insufficient, and strain dynamics would likely also need to be incorporated. Particularly because our main focus here is on the interaction parameters, and not on immunity dynamics over time, we chose to keep the model as simple as possible, and to avoid introducing additional unknown parameters. We have now better justified this decision in the manuscript as well (lines 537-538).

ll 322–324: Similarly, in this passage, the choice to make the fraction of the population susceptible at the beginning of the season is presented as an alternative to Waterlow’s decision to make this a constant. Of course, it is an alternative, and a more reasonable one, but it is not the only alternative, nor is it the most intuitive one. Why do the authors take this approach?

One way to address this would be to discuss the observed differences in season-specific parameters from one season to the next (Supp Fig 3) and to plot these against estimates of the total number of within-season infections. In other words, how does the estimate of S at the beginning of a season depend on the total number of infections in the preceding season?

Hopefully this has been partially answered by our response to your previous point - by allowing the model to fit the proportion immune at the beginning of each season, we avoid having to explicitly model processes such as the loss of immunity over time. Our responses to some of Reviewer 2’s comments are probably also relevant here. Specifically, given the inherent weaknesses of surveillance data, it is impossible to calculate the true attack rate for a given season; for this reason, it is difficult to base the proportion immune at the beginning of a season on the previous season’s attack rate. Therefore, we allow the model to fit these values, so that it can determine which values are most consistent with the observed data for each individual season. We now explain this in our discussion (lines 378-387).

That said, it is a good idea to at least check whether the inferred proportion immune at the beginning of the season is higher after seasons where there were more cases, as we would tend to expect. We now show the relationship between the proportion immune at the beginning of each season and the estimated attack rate or observed number of cases from the previous season in Supplementary Fig. 5. Encouragingly, we see that, as the estimated influenza attack rate and the total number of observed influenza cases from the previous season increases, the inferred value for the proportion immune to influenza at the season’s start also tends to increase, suggesting that the fit values are, at least to some extent, accounting for immunity remaining from the previous season. Very little pattern is observed for RSV, although this could be because RSV attack rates are fairly uniform from season to season.

l 195ff & Fig 3: The R^2 values quoted amount to a comparison of the model's predictions relative to those of a model in which the cases are constant through time. It would be informative to similarly present a comparison with a more descriptive model, such as one which captures the average seasonality.

In addition to comparing the model predictions to the average observed cases over time, we now also compare the model predictions to the values obtained by fitting a sine wave to the observed data. Specifically, we fit a sine wave through the entire time series for a given location, rather than fitting separate sine waves to each individual season. We then recalculated the coefficients of efficiency (R^2), this time comparing the fit of our model in the main text to the fits from the sine waves (such that positive values indicate that the mechanistic model fits better, and negative values indicate that the sine wave fits better). Our model offers a similar improvement in fit over the sine waves as over the average observed cases, except in the case of RSV in Canada, where our model and the sine wave fit similarly ($R^2 = 0.90$ and 0.53 for influenza and RSV, respectively, in Hong Kong; $R^2 = 0.91$ and -0.04 for influenza and RSV, respectively, in Canada; these values can also now be found in Supplementary Table 3). These comparisons are now described in lines 204-206.

Supp Fig 6: The profile likelihoods could be solidified. More exploration is clearly needed at very low values of these parameters and it may be necessary, for example, to use a logarithmic horizontal scale. Also, a proper profile likelihood plot should have a vertical scale showing at most 10 or 20 log units—at least if the Wilks theorem applies—since one log unit of resolution is then meaningful.

We agree, and have performed fits holding $\theta_{\lambda 1}$ to values from 0 to 0.02 in Hong Kong. We now present the results of these fits in Supplementary Fig. 10, rather than the previous profile likelihoods using values from 0 to 0.2. Values above 0.011 (in Hong Kong) consistently yielded log-likelihoods that were more than 20 points lower than the maximum log-likelihood values and were therefore not plotted in order to keep the y-axis to a more meaningful range. Although the profile likelihoods still appear a bit noisy, we feel that some noise is expected due to the high dimension of the problem (i.e., 40-54 parameters fit).

Finally, the variability in the estimates for the Influenza B-RSV interaction cast doubt on the identification of the MLE.

Rather than necessarily casting doubt on the identification of the MLE, the wide confidence intervals for the influenza B-RSV analysis suggest that the data do not contain enough information to allow the parameters of interest to be inferred (in other words, changes to the values of the parameters of interest have little influence on the log-likelihood values). As described above, we have removed the subtype-specific analyses and have rerun our model fitting procedure using the sum of H1N1 and B influenza in Hong Kong, which allows for the parameters of interest to be identified.

Minor remarks

l 48: I believe “host cell” should be hyphenated here.

We have updated the text to read “host-cell.”

l 50: It would be better not to begin two successive paragraphs with the same word.

Agreed - this has been changed.

l 76ff: As written, this sentence is awkward. I suggest recasting as “It is, as usual, unclear whether

and how results from studies in animal models are applicable to human populations. Moreover, seemingly intuitive study designs, ...”

We have now split this sentence up to improve readability.

I 78: What is meant by “phase differences”? Is this meant to describe differences in the timing of epidemic peaks between different infections? As it stands, this is unclear.

Exactly - by phase differences, we mean the difference in timing of the two pathogens. We have added this definition to the text (line 78).

I 81: It is unclear what is meant by the “large number of components” needed to characterize interactions. Please clarify.

We have removed this sentence, as we agree that it was confusing rather than helpful.

II 82–83: Mathematical models are useful not only because they can express complex and nonlinear dynamics and account for various mechanisms, but because they establish logical linkages between these things and observable data.

This is true, and helps to better convey why these qualities of models are so helpful. We have added a sentence to communicate this (lines 83-85).

I 137: The point estimate is reported to 3 significant figures, but the uncertainty does not support this. Perhaps a formulation like “...reduction in susceptibility persists for roughly 100 days (point estimate 111 d, 95% CI 96–127 d)...” would be better. There are other instances of similarly overprecise statements.

Thank you for calling attention to this. We have gone through the manuscript and attempted to address any values reported with too much precision.

I 148: Given that the duration of potential effect of RSV on Influenza B is not identifiable, it is not surprising that the magnitude is similarly unidentified. Indeed, it would be strange if it were otherwise, no?

This is true - it would have been odd to find a specific value for the interaction strength but a wide range of values for the duration. Since we removed the subtype-specific analyses, the original statement is no longer included.

II 165, 171: “[C]ontribute significantly” is not the right language in this context. It would be better to say that temperature and absolute humidity both “modulate transmission”. Similarly, in I 171, temperature and humidity cannot be spoken of as “drivers” of transmission, but rather as “modulators”.

We have updated the wording in these lines accordingly.

I 180–181: The mathematical symbols have not yet been defined and use unfamiliar notation. In particular, notation such as R_{20} is annoyingly idiosyncratic. It would be preferable to write R_{20} .

A description of these parameters has been added, and we have updated the notation as suggested.

Make sure all mathematical symbols are defined in the text as they are first used (e.g., I 148, II 180–181).

θ_{λ_2} is now defined on line 152. A description has also been added for the parameters introduced in lines 180–181 (now lines 182–186).

I 285–286: It would be better to say “...we found evidence for an interaction effect...”

We have now reworded as suggested.

II 351–352: “...we expect that all influenza (sub)types will be influenced by climate to some extent.” Indeed, this is consistent with your findings, as would be worth reminding us here.

As this statement concerned a comparison between the results of the model fit using influenza H1N1 data and the model fit using influenza B data, it has been removed.

II 428–429: “...a previous modeling study found that the impact of influenza on susceptibility to RSV could not be inferred using data from Vietnam.” The reader naturally wonders why the earlier study failed. Some explanation, or even speculation, would be appreciated here.

Again, since this statement was made to compare the study in Vietnam to the results of our model fit to influenza B data, it has been removed now that we no longer perform subtype-specific analyses. However, the referenced paper is discussed more in lines 328–351; in particular, we highlight where our work and this past study differ in the key assumptions made and data used, and how this may have contributed to differences in the fit interaction parameters. Since we also found that there were differences in the strength of the interaction effect as estimated in Hong Kong vs. Canada, we also note that some of the differences in results between the two manuscripts could simply be due to a difficulty in inferring consistent interaction characteristics when fitting to multiple, individual locations (lines 349–351).

II 446–448: “This suggests that homogeneous models are sufficient to capture the broad characteristics of this interaction, even when the underlying transmission dynamics vary by age group...” Unless I am mistaken, this will not be true in general. Indeed, if the coupling between age groups is weak, there may be substantial differences between the predictions of an age-structured and a homogeneous model. As written, this conclusion appears more general than I believe the authors may intend.

Thank you for noticing this. We have revised the sentence to clarify that we are speaking about the applicability of homogenous models for the particular case considered in this paper, and not in general.

I 596: There appears to be a typo in Eqn 2.

Thank you for noticing this. We have corrected the equation.

Reviewers' Comments:

Reviewer #1:

Remarks to the Author:

Thank you for the extensive work you have done in response to the previous reports. The manuscript is informative and important.

You say in your conclusions (line 500 etc.): "infection with either influenza virus or RSV causes a moderate to strong reduction in susceptibility to the other virus for up to several months, and that this individual-level effect can lead to substantial ramifications at the population level. We further explore the implications of this finding for widespread LAIV use, and find that the effect of vaccination with LAIV at the population level is highly dependent on the underlying transmission dynamics of both viruses in a given location and season".

Might you add some actual numbers saying how large the effects are and how long they last? You do give a wide statement of size and duration of effect in the Abstract, but might you add more details?

Reviewer #2:

Remarks to the Author:

The authors have gone through substantial efforts to address my concerns and those of the other reviewers. Thank you for that. The arguments provided and the additional analyses are largely convincing. Though some scepticism remains on my end, and hence I am also glad to see that the authors have moderated their claims on the strength of the provided evidence to convey that it adds to a large base of often conflicting evidence.

Reviewer #3:

Remarks to the Author:

At the forefront of scientific research, there are always a great many choices to be made without unambiguous guidance. If there is to be progress in science, it must be possible to make reasonable choices, explore their consequences, and report the results judiciously. The authors have done a very thorough exploration of the issues here. Their choices appear to have been carefully made and thoughtfully discussed. The conclusions are not over-interpreted or over-sold. In particular, they have addressed all the concerns I raised in the first review.

I recommend that the paper be accepted for publication forthwith.

Reviewer #4:

None

REVIEWER COMMENTS

Reviewer #1 (Remarks to the Author):

Thank you for the extensive work you have done in response to the previous reports. The manuscript is informative and important.

Thank you.

You say in your conclusions (line 500 etc.): "infection with either influenza virus or RSV causes a moderate to strong reduction in susceptibility to the other virus for up to several months, and that this individual-level effect can lead to substantial ramifications at the population level. We further explore the implications of this finding for widespread LAIV use, and find that the effect of vaccination with LAIV at the population level is highly dependent on the underlying transmission dynamics of both viruses in a given location and season".

Might you add some actual numbers saying how large the effects are and how long they last? You do give a wide statement of size and duration of effect in the Abstract, but might you add more details?

We have added quantitative information on the estimated strength and duration of the interaction effect to our conclusions, and have furthermore been more specific about the potential impacts at the population level (lines 500-503). Because our analysis of the impacts of LAIV use relies on several assumptions, we have decided to keep our closing statement on these results purely qualitative, and encourage readers to focus on the overall patterns found, rather than the specific quantitative results (see lines 402-409).

Reviewer #2 (Remarks to the Author):

The authors have gone through substantial efforts to address my concerns and those of the other reviewers. Thank you for that. The arguments provided and the additional analyses are largely convincing. Though some scepticism remains on my end, and hence I am also glad to see that the authors have moderated their claims on the strength of the provided evidence to convey that it adds to a large base of often conflicting evidence.

Thank you, and we are glad that we were able to improve the manuscript to make our work more convincing.

Reviewer #3 (Remarks to the Author):

At the forefront of scientific research, there are always a great many choices to be made without unambiguous guidance. If there is to be progress in science, it must be possible to make reasonable choices, explore their consequences, and report the results judiciously. The authors have done a very thorough exploration of the issues here. Their choices appear to have been carefully made and thoughtfully discussed. The conclusions are not over-interpreted or over-sold. In particular, they have addressed all the concerns I raised in the first review.

I recommend that the paper be accepted for publication forthwith.

Thank you for your thoughtful and encouraging remarks.